# One-step synthesis of sequence-controlled multiblock polymers with up to 11 segments from monomer mixture

Xiaochao Xia [1,2✉], Ryota Suzuki[3], Tianle Gao[3], Takuya Isono [2✉] & Toshifumi Satoh [2✉]

Switchable polymerization holds considerable potential for the synthesis of highly sequence-controlled multiblock. To date, this method has been limited to three-component systems, which enables the straightforward synthesis of multiblock polymers with less than five blocks. Herein, we report a self-switchable polymerization enabled by simple alkali metal carboxylate catalysts that directly polymerize six-component mixtures into multiblock polymers consisting of up to 11 blocks. Without an external trigger, the catalyst polymerization spontaneously connects five catalytic cycles in an orderly manner, involving four anhydride/epoxide ring-opening copolymerizations and one L-lactide ring-opening polymerization, creating a one-step synthetic pathway. Following this autotandem catalysis, reasonable combinations of different catalytic cycles allow the direct preparation of diverse, sequence-controlled, multiblock copolymers even containing various hyperbranched architectures. This method shows considerable promise in the synthesis of sequentially and architecturally complex polymers, with high monomer sequence control that provides the potential for designing materials.

[1] College of Materials Science and Engineering, Chongqing University of Technology, Chongqing 400054, China. [2] Division of Applied Chemistry, Faculty of Engineering, Hokkaido University, Sapporo 060-8628, Japan. [3] Graduate School of Chemical Sciences and Engineering, Hokkaido University, Sapporo 060-8628, Japan. ✉email: xiaxiaoc@cqut.edu.cn; isono.t@eng.hokudai.ac.jp; satoh@eng.hokudai.ac.jp

Copolymers are long macromolecular chains composed of at least two monomers of different chemical natures. High monomer-sequence regulation enables effective control of structure–property relations of copolymers so that precise sequence-controlled polymers may be endowed with novel properties or functions[1–3]. In this context, considerable efforts have been made on the development of various synthetic methods, including "click" reactions[4,5], sequential monomer addition[6–8], and solid-phase synthesis[9–11]. Although these strategies have made significant progress for the synthesis of sequence-controlled block polymers, they are hampered by disadvantages, such as being extremely complex and time-consuming, as well as iterative monomer attachment/deprotection. This increases costs and often leads to poor yields, thereby making it challenging to expand their application[12–14].

"Switchable polymerization" has been exploited for the spontaneous, selective transformation of a monomer mixture into a sequence-controlled block copolymer in one synthetic step, thereby overcoming the disadvantages of conventional procedures[15–29]. Such a catalytic system bridges two catalytic cycles among ring-opening copolymerization (ROCOP) of anhydrides or carbon dioxides/epoxides and ring-opening polymerization (ROP) of cyclic esters or epoxides (Fig. 1a)[15–17]. Therefore, switchable polymerization shows unprecedented advantages, including applicability to numerous commercially available, industrially relevant monomers[30–33], and enabling the synthesis of diverse, functional, and sequence-controlled multiblock polymers[26,34,35]. Switchable catalysis has been applied to various metal-complex catalysts and organocatalysts, where the initial anhydride or carbon dioxide/epoxide ROCOPs were followed by the ROPs of cyclic esters, forming various block polyesters[15–17,25,27,36–39]. Most of these prior studies were mainly limited to di- or triblock copolymers, and one-step preparation of longer block sequences was rarely performed. Recently, Williams et al. developed a cocatalyst system (a Cr(III) salen complex, [SalcyCrCl]; bis(triphenylphosphoranylidene)ammonium chloride, PPNCl) that exploited mechanistic switches between an anhydride/epoxide ROCOP, epoxide ROP, and a lactone ROP, resulting in the one-step synthesis of ABCBA-type pentablock terpolymers[26]. Although this catalytic system has shown extraordinary performance in the synthesis of well-controlled multiblock polymers, it remains limited to a three-component system.

Previously, we explored alkali-metal carboxylate catalysts to develop a self-switchable polymerization system for the copolymerization of epoxides, cyclic anhydrides, and cyclic esters. This green, biocompatible, low-cost catalytic system spontaneously combined epoxide/cyclic anhydride ROCOP and cyclic ester ROP to produce various well-defined multiblock polyesters[35]. In this study, the self-switchable polymerization is expanded from a three- to a six-component system, wherein the catalyst spontaneously links five catalytic cycles, involving four ROCOPs of anhydrides/epoxides and one ROP of cyclic esters. This results in a one-step synthetic pathway for preparing multiblock polymers consisting of up to 11 blocks (Fig. 1b). Conventional switchable catalysis indicates a high kinetic favorability for anhydride/epoxide ROCOP relative to lactide (LA) ROP, resulting in the ROCOPs of anhydrides/epoxides occurring prior to the ROPs of cyclic esters. In our catalytic system, the polymerization order is manipulated by facile-controlling the differences in reactivities between LA and cyclic anhydrides (Fig. 1b).

## Results
### Polymerization of two anhydrides and an epoxide. 
Initially, ROCOP within a mixture of diglycolic anhydride (DGA) and 5-norbornene-endo-2,3-dicarboxylic anhydride (NA) with ethyl glycidyl ether (EGE) was performed using cesium pivalate (*t*-BuCO₂Cs) and 1,4-benzenedimethanol (BDM) as the catalyst and bidirectional initiator, respectively, to evaluate the possibility of sequential incorporation. The initial attempt led to a sequence-defined, ABA-type, triblock copolymer rather than a random copolymer. Initially, the ROCOP of DGA/EGE occurs, as shown by the decrease in the $^1$H nuclear magnetic resonance (NMR) signals representing DGA at 4.43 ppm (Supplementary Fig. 1, proton 3′) and appearance of $^1$H NMR signals representing P(DGA-*alt*-EGE) at 5.29−5.20 (Supplementary Fig. 1, proton 5) and 4.27−4.22 ppm (Supplementary Fig. 1, proton 3). Concurrently, the peak at 6.37−6.13 ppm (Supplementary Fig. 1, proton 12) does not appear, and the signal at 6.30 ppm (Supplementary Fig. 1, proton 12′ of NA) remains unchanged, indicating that NA is unreacted, without any trace of P(NA-*alt*-EGE) formation. The ROCOP of NA/EGE commences after DGA is completely consumed (3.7 h, % DGA Conv. > 99; % NA Conv. = 0), finally forming P(NA-*alt*-EGE)-*b*-P(DGA-*alt*-EGE)-*b*-P(NA-*alt*-EGE) triblock copolymers (entry 1 in Table 1 and Supplementary Figs. 1 and 2), and signals associated with both block sequences were observed in $^1$H, $^{13}$C, and 2D NMR spectra (Supplementary Figs. 3–5). The formation of triblock copolymers rather than blends of P(NA-*alt*-EGE) and P(DGA-*alt*-EGE) is further demonstrated using diffusion-ordered NMR spectroscopy (DOSY), which shows only one diffusion coefficient (Supplementary Fig. 6). Notably, no P(NA-*alt*-EGE) is inserted into the P(DGA-*alt*-EGE) block, even though the former forms significantly faster than the latter. Size-exclusion chromatography (SEC) reveals a monomodal molecular weight distribution ($Đ = 1.37$, entry 1 in Table 1 and Supplementary Fig. 7). Therefore, the polymerization system shows an ideal chemoselectivity.

### Mechanistic studies. 
In our previous study, the perfect alternating structure of copolymers from ROCOP has been confirmed by using $^1$H NMR, $^{13}$C NMR, and MALDI-TOF MS characterization. The reactivity difference among anhydride, epoxide, and cyclic ester and a multiactivation mechanism in an alkali metal carboxylate catalytic system has been illustrated in detail[35]. Supported by these, a chemoselective mechanistic pathway for terpolymerization from the mixture of DGA/NA/EGE is proposed in Fig. 2. Because DGA is far more reactive than NA, as shown by the $^1$H NMR spectra in Supplementary Fig. 8, the activated DGA is more prone to nucleophilic attack from the cesium pivalate-activated hydroxyl group relative to the activated NA. The resulting carboxylate species from the ring opening of DGA reacts with the cesium cation-activated EGE, forming a copolymer with an accurately alternating chemical structure. When the DGA is completely consumed, the termini of P(DGA-*alt*-EGE) are occupied by hydroxyl groups, eventually inducing the ROCOP of NA/EGE in a similar manner and forming a triblock copolymer. The nonterminal model yielded reactivity ratios of $r_{DGA} = 827.46 \pm 50.30$ and $r_{NA} = 0.0036 \pm 0.0006$ (Supplementary Fig. 9). On the basis of these reactivity ratios, we concluded that the resultant polymers were most consistent with a nearly perfect triblock copolymer. Due to $k_1 \gg k_4$ which was reported by Coates and Williams et al.[22,35,40–42], the slow insertion of epoxides is the rate-determining step for polyester formation (Supplementary Fig. 10).

### Combinations of various ROCOPs. 
Based on the initial findings, a series of experiments were conducted to evaluate the reactivity ratio of the anhydrides (Supplementary Figs. 8, 9, and 11–14), and the reactivity trend determined as DGA ≫ SA ≫ NA ≫ DPMA (Supplementary Fig. 15). Hence, the length of sequence-defined multiblock polymers can be simply and flexibly adjusted by controlling the number of catalytic cycles in the self-switchable

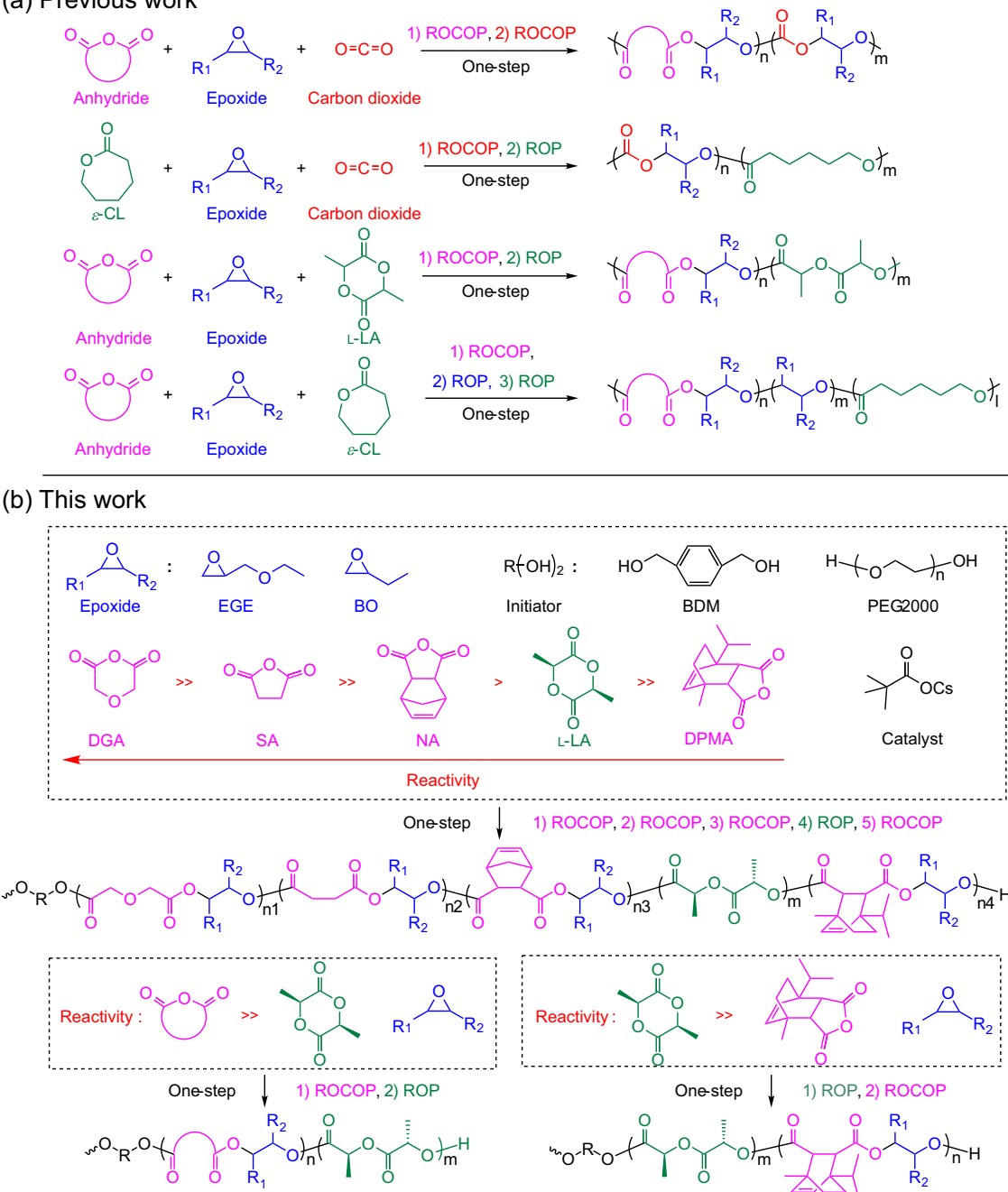

**Fig. 1 One-step synthesis of multiblock polymers from a monomer mixture. a** Previous strategies are only applicable to three-component monomer mixture. **b** The strategy presented in this paper provides multiblock polymers from six-component monomer mixture. L-LA L-lactide, DGA diglycolic anhydride, NA 5-norbornene-endo-2,3-dicarboxylic anhydride, EGE ethyl glycidyl ether, SA succinic anhydride, DPMA rac-cis-endo-1-isopropyl-4-methyl-bicyclo[2.2.2]oct-5-ene-2,3-dicarboxylic anhydride, BO 1,2-butylene oxide, ROCOP ring-opening copolymerization, ROP ring-opening polymerization, PEG2000 polyethylene glycol (molecular weight = 2 kDa). Pink: anhydride. Red: carbon dioxide. Blue: epoxide. Green: cyclic ester.

polymerization, thereby overcoming the limitations of di- or triblock copolymers (Fig. 3). This is demonstrated by polymerization within a mixture of DGA/SA/NA/DPMA/BO with a BDM initiator, which produces heptablock polymers via a one-step procedure. Based on the evolution of the $^{1}$H NMR spectra (Supplementary Fig. 16), the reaction progresses in four stages: first, DGA is consumed with increasing reaction time and is finally completely consumed, as shown by the disappearance of the $^{1}$H NMR signals at 4.43 ppm representing DGA, while the broad signals representing the P(DGA-*alt*-BO) block appear at 5.11−5.02 (proton 5, Supplementary Fig. 16) and 4.27−4.22 ppm

(proton 3, Supplementary Fig. 16). After complete consumption of DGA, the $^{1}$H NMR signals at 3.01 ppm representing SA begin to decrease, reaching at 2.69−2.58 ppm (proton 8, Supplementary Fig. 16) representing P(SA-*alt*-BO). This indicates the commencement of ROCOP of SA/BO, thereby forming the P(SA-*alt*-BO) block. Upon almost 99% conversion of SA, the ROCOP of NA/BO consumes NA, which is confirmed by the $^{1}$H NMR signals diminishing at 6.30 ppm representing NA and instead occurring at 6.37−6.13 ppm (proton 14, Supplementary Fig. 16) representing P(NA-*alt*-BO). After NA is completely consumed, the distinct doublets at 6.08 and 6.00 ppm representing the

**Table 1 The self-switchable polymerizations catalyzed by cesium pivalate[a].**

| entry | monomers | $[\text{anhydride}]_0/[\text{L-LA}]_0/[\text{epoxide}]_0/[\text{BDM}]_0/[\text{cat.}]_0$ | Temp. [°C] | Time (h) | conv.[b] [%] (anhydride or L-LA) | $M_{n,th.}$[c] [kDa] | $M_{n,NMR}$[b] [kDa] | $M_{n,SEC}$[d] [kDa] | Đ[d] |
|---|---|---|---|---|---|---|---|---|---|
| 1 | DGA/NA/EGE | 25/25/150/2/1 | 100 | 6.5 | DGA > 99, NA = 92 | 5.9 | 6.8 | 5.8 | 1.37 |
| 2 | DGA/SA/NA/DPMA/BO | 25/25/25/12.5/250/2/1 | 100 | 73 | DGA > 99, SA > 99, NA > 99, DPMA = 86 | 9.2 | 9.8 | 7.5 | 1.31 |
| 3 | DGA/SA/NA/EGE | 25/25/25/250/2/1 | 100 | 7 | DGA > 99, SA > 99, NA = 71 | 7.7 | 7.2 | 6.9 | 1.28 |
| 4 | DPMA/L-LA/EGE | 25/50/250/2/1 | 100 | 23 | L-LA > 99, DPMA = 80 | 6.7 | 7.2 | 2.0 | 1.51 |
| 5 | DPMA/BO | 25/125/2/1 | 80 | 41 | DPMA > 99 | 3.9 | n.d. | 2.1 | 1.16 |
| 6 | DPMA/L-LA/BO | 25/50/250/2/1 | 80 | 41 | L-LA > 99, DPMA = 45 | 5.3 | 6.4 | 4.2 | 1.35 |
| 7 | DGA/SA/NA/L-LA/DPMA/BO | 25/25/75/12.5/350/2/1 | 100 | 97 | DGA > 99, SA > 99, NA > 99, L-LA > 99, DPMA = 86 | 14.6 | 14.9 | 6.6 | 1.51 |
| 8 | DGA/SA/NA/L-LA/DPMA/BO | 25/25/75/12.5/350/2/1 | 80 | 190 | DGA > 99, SA > 99, NA > 99, L-LA > 99, DPMA = 76 | 14.4 | 15.4 | 8.0 | 1.41 |
| 9 | DGA/SA/L-LA/DPMA/BO | 25/25/75/350/2/1 | 100 | 97 | DGA > 99, SA > 99, L-LA > 99, DPMA = 41 | 10.8 | 11.1 | 6.8 | 1.23 |
| 10[e] | DGA/SA/NA/L-LA/DPMA/BO | 25/25/75/12.5/350/2/1 | 80 | 66 | DGA > 99, SA > 99, NA > 99, L-LA > 99, DPMA = 91 | 16.5 | 15.5 | 7.8[d], 18.7[f] | 1.46 |
| 11 | TA/NA/L-LA/DPMA/BO | 25/25/50/12.5/350/2/1 | 80 | 24 | TA > 99, NA > 99, L-LA > 99, DPMA = 54.5 | 11.9 | 12.2 | 7.7 | 1.60 |
| 12 | TA/NA /BO | 25/25/150/2/1 | 80 | 10 | TA > 99, NA = 60.5 | 6.0 | 5.8 | 4.3 | 1.61 |
| 13 | TA/L-LA/DPMA/BO | 25/50/12.5/350/2/1 | 80 | 25 | TA > 99, L-LA > 99, DPMA = 50 | 8.9 | 9.3 | 5.1 | 1.71 |

[a]Polymerization conditions: Ar atmosphere.
[b]Determined by $^1$H NMR analysis of the obtained polymer in CDCl$_3$.
[c]Theoretical $M_n$ values.
[d]Determined by the SEC analysis of the obtained polymer in THF with a PSt standard.
[e]PEG2000 is used as a bidirectional initiator.
[f]Determined by SEC with multi-angle light scattering detector (SEC-MALS).

disappeared DPMA, are replaced by a broad signal at 6.12–5.96 ppm (protons 20 and 21, Supplementary Fig. 16) due to the formation of the P(DPMA-*alt*-BO) block by the ROCOP of DPMA/BO (Supplementary Figs. 16–19). This catalysis, involving a five-component mixture, remains highly selective, almost without the tapered region[14,43,44], as clearly indicated by the plots of monomer conversion as a function of time (Supplementary Fig. 17). The SEC trace of the resulting polymer exhibits a well-defined monomodal molecular weight distribution (Đ) and the molecular weight increases continuously with monomer consumption, accompanied by a narrow, unimodal distribution (entry 2 in Table 1 and Supplementary Fig. 20), and DOSY reveals only one diffusion coefficient (Supplementary Fig. 21), further demonstrating a single heptablock polymer formation rather than a blend[8,45,46]. Using this method, a well-defined pentablock polymer is also successfully obtained by polymerizing a mixture of DGA/SA/NA/EGE, which also displays a high level of monomer-sequence control (entry 3 in Table 1, Supplementary Tables 1 and 2, and Supplementary Figs. 22–27).

**Polymerization of DPMA, EGE, and L-LA.** To further increase the diversity of multiblock polymers, the reactivities of DPMA and L-LA were studied by incorporating L-LA into the system through polymerization within a mixture of DPMA/EGE. Initially, we expected the polymerization to follow a conventional pathway (Fig. 4a), wherein the initial anhydride/epoxide ROCOP is followed by the ROP of L-LA, with propagation occurring from the termini of the resultant alternating copolymer, finally forming PLLA-*b*-P(DPMA-*alt*-EGE)-*b*-PLLA triblock copolymers[21,28]. However, the evolution of the $^1$H NMR spectra reveals a unique propagation pathway (Fig. 4b, entry 4 in Table 1, and Supplementary Figs. 28 and 29). The ROP of L-LA occurs first, as shown by the decrease in the $^1$H NMR signals at 5.08 ppm representing L-LA and reaching at 5.27–5.09 ppm (proton 3, Supplementary Fig. 28) representing PLLA. During this period, no broad peak is observed at 6.12–5.96 ppm (protons 8 and 9, Supplementary Fig. 28), and the signals at 6.08 and 6.00 ppm representing DPMA remain unchanged, indicating that DPMA remains unreacted, with no P(DPMA-*alt*-EGE) block formation. The ROCOP of DPMA/EGE only commences after L-LA is completely consumed, finally producing a P(DPMA-*alt*-EGE)-*b*-PLLA-*b*-P(DPMA-*alt*-EGE) triblock copolymer (Supplementary Figs. 28–32). The nonterminal model yielded reactivity ratios (Supplementary Fig. 33, $r_{L-LA} = 831.91 \pm 49.22$ and $r_{DPMA} = 0.0069 \pm 0.0002$), which confirmed that the perfect triblock copolymer should be formed. The analysis of SEC showed that the resultant copolymer shows ill-defined features, such as a broad monomodal Đ (1.51, entry 4 in Table 1 and Supplementary Fig. 31). This is due to that transesterification or back-biting reactions could occur during polymerization for DPMA/L-LA/EGE system. ROCOP of DPMA/BO provided a well-defined copolymer with monomodal molecular weight distribution and a narrow Đ (1.16, entry 5 in Table 1 and Supplementary Figs. 34–36). The polymerization from the mixture of DPMA/L-LA/BO was carried out at 80 °C, and the resultant triblock polymer displayed the better controllability (Đ = 1.35, entry 6 in Table 1 and Supplementary Figs. 37–39) relative to DPMA/L-LA/EGE system. This result indicated that side reactions such as transesterification or back-biting reactions can be suppressed by lowering reaction temperature. The polymerization system displays an unprecedented propagation pathway stemming from the far-higher reactivity of L-LA than that of DPMA, with the very low reactivity of DPMA attributed to steric hindrance. Therefore, the polymerization order can be freely manipulated between anhydride/epoxide ROCOP and L-LA ROP using differences in the reactivities of the monomers (Fig. 4).

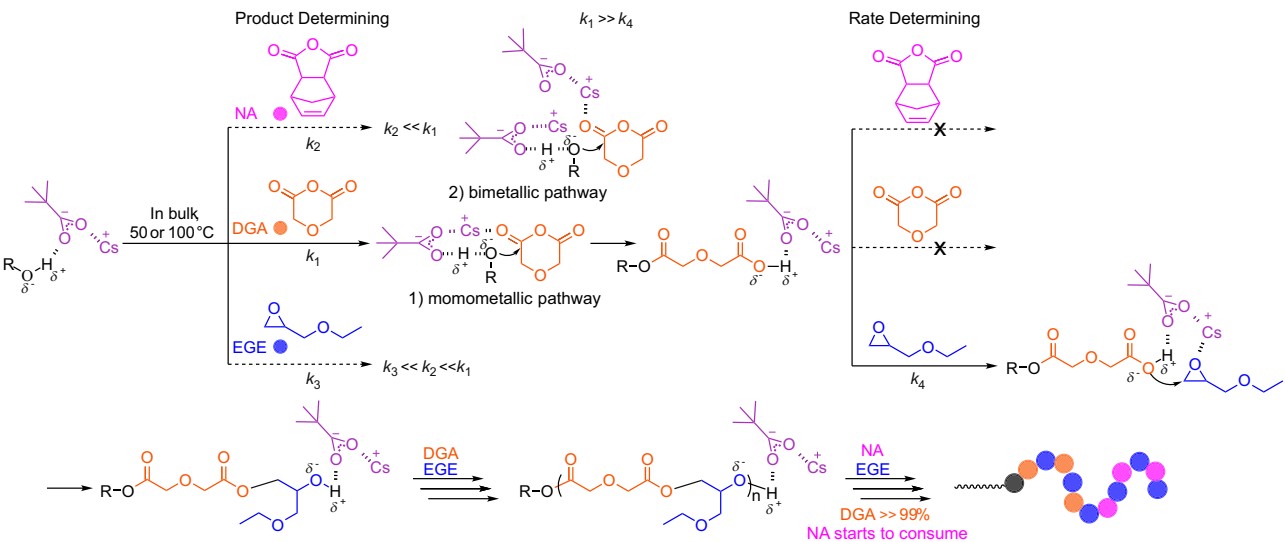

**Fig. 2 Proposed mechanism of self-switchable polymerization from a mixture of DGA, NA, and epoxides with cesium pivalate as the catalyst and alcohol as the initiator.** Possible mechanism in the ring-opening step includes momometallic pathway and bimetallic pathway. Pink: NA. Orange: DGA. Blue: EGE.

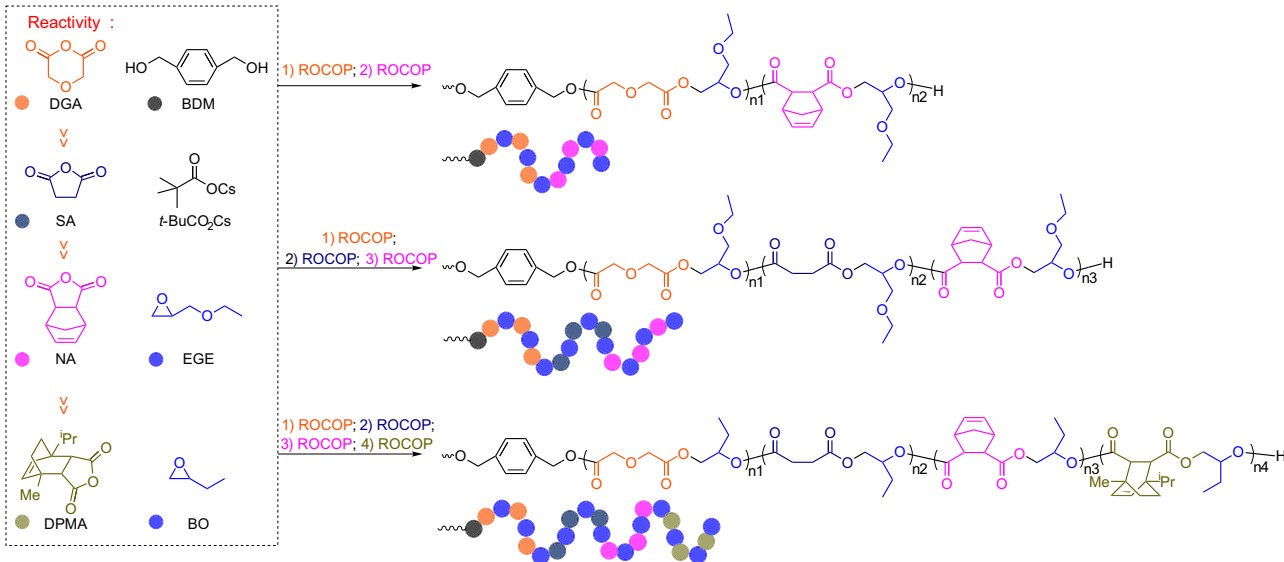

**Fig. 3 The synthesis of multiblock polymers with various sequences by the combination among different anhydride/epoxide ROCOP at 100 °C.** Orange: DGA. Pink: NA. Dark blue: SA. Blue: EGE or BO. Olive: DPMA.

Combining these results with our previous work[35], where the ROCOP of NA/epoxide was kinetically preferred over L-LA ROP, the reactivity trend is DGA ≫ SA ≫ NA > L-LA ≫ DPMA. Therefore, we used cesium pivalate as a catalyst to properly link five catalytic cycles, involving four ROCOPs of anhydrides/epoxides and one ROP of L-LA, to prepare sequence-defined multiblock polymers consisting of up to nine blocks. The simultaneous polymerization of four different anhydrides, BO, and L-LA was monitored via the evolution of the $^1$H NMR spectrum (Fig. 5). Initially, DGA/BO ROCOP forms a polyester (Supplementary Fig. 40), followed by the serial incorporation of P(SA-*alt*-BO) and P(NA-*alt*-BO) blocks generated by SA/BO ROCOP and NA/BO ROCOP, respectively (Supplementary Figs. 41 and 42). After 28.7% conversion of NA, L-LA also commences reacting, leading to a clearly tapered region (the molar mass fraction of tapered region is 67.7%). This tapered region is reduced by lowering the reaction temperature from 100 to 80 °C (Figs. 6a and 6b), due to

the increasing difference in the reactivities of L-LA and NA with lowering temperature (Supplementary Figs. 43 and 44). Upon 99% conversion of L-LA (Supplementary Fig. 45), slow consumption of DPMA by ROCOP of DPMA/BO occurs, finally producing a multiblock polymer consisting of up to nine blocks. The formation process of the multiblock polymer was also confirmed by the evolution of the $^{13}$C NMR spectrum (Supplementary Fig. 46). In particular, the $^{13}$C NMR signals associated with carbonyl groups of DGA (carbon 1′), SA (carbon 7′), and NA (carbon 11′) disappeared one by one, while the $^{13}$C NMR signals associated with carbonyl groups of P(DGA-*alt*-BO) (carbon 1), P(SA-*alt*-BO) (carbon 7), and P(NA-*alt*-BO) (carbon 11) segments appeared in succession. After 43.9% conversion of NA, the $^{13}$C NMR signals associated with carbonyl groups of L-LA (carbon 18′) also commenced decreasing, accompanied by the appearance of the $^{13}$C NMR signals associated with carbonyl groups of the PLLA segment (carbon 18). During the process, the

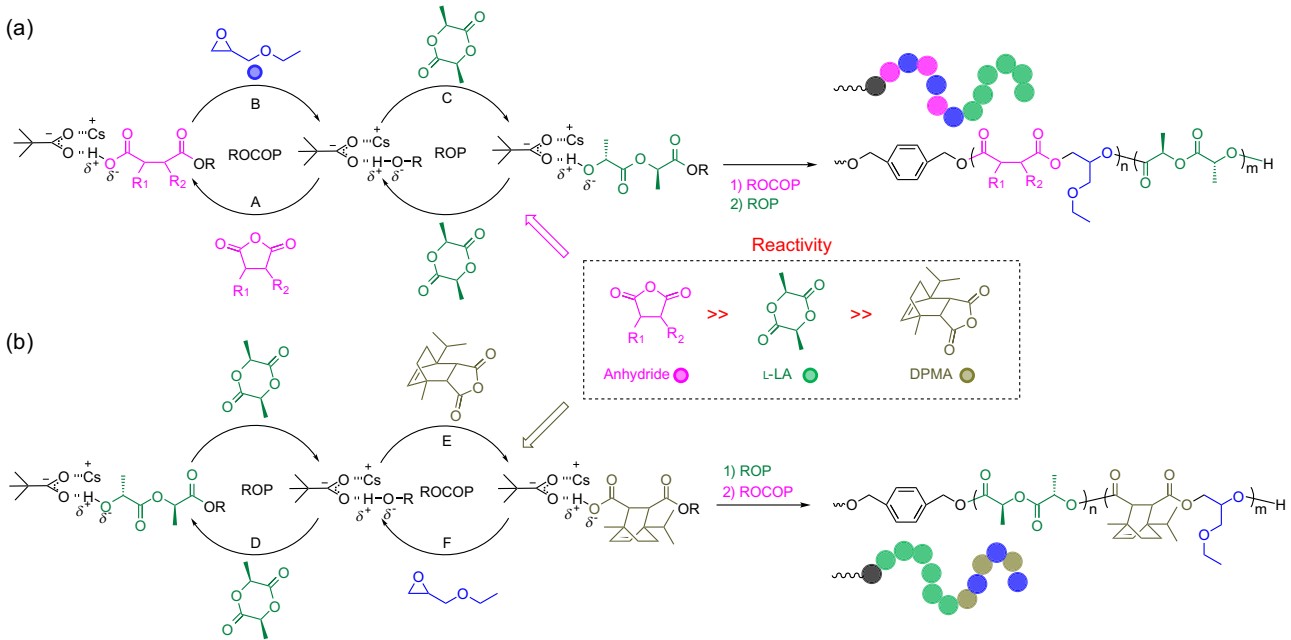

**Fig. 4 Two different polymerization routes of the mixture of anhydride, epoxide, and ʟ-LA. a** The conventional route. **b** The present route. Green: ʟ-LA. Blue: EGE. Olive: DPMA.

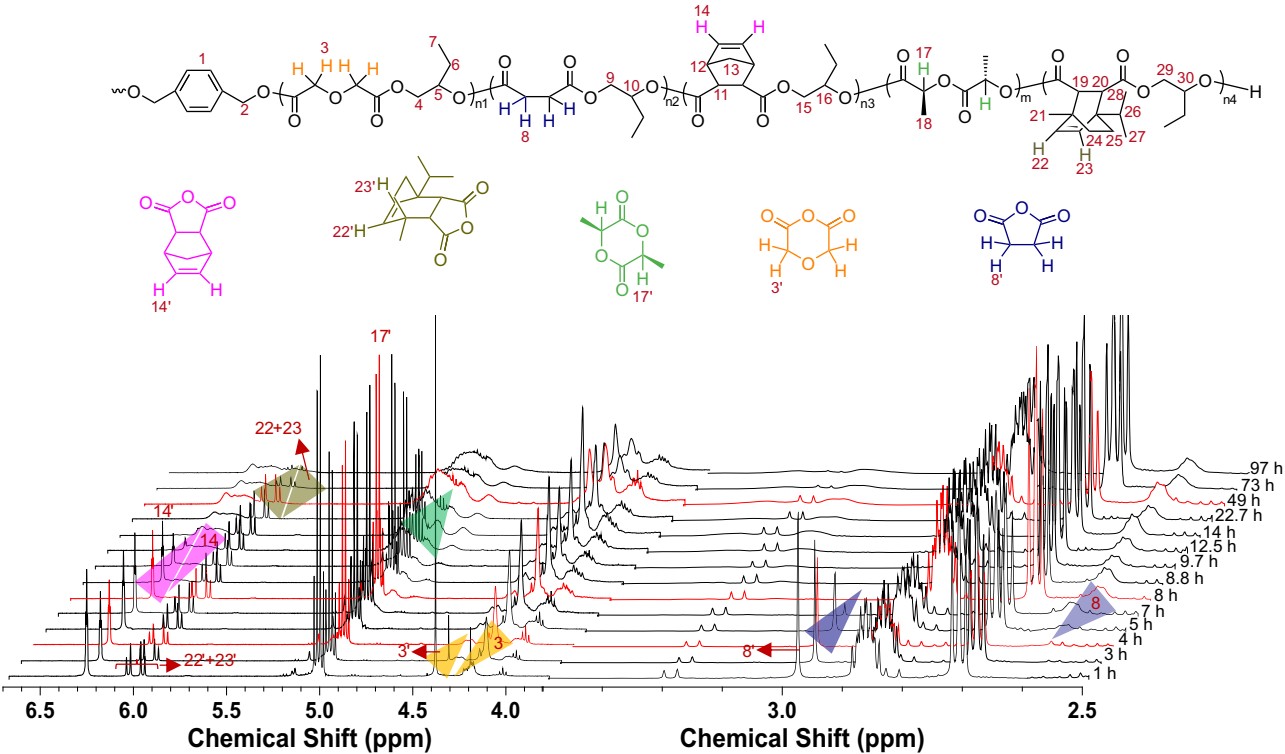

**Fig. 5 The ¹H NMR (CDCl₃) spectra analysis of the self-switchable polymerization from the mixture of DGA/SA/NA/ʟ-LA/DPMA/BO.** The ¹H NMR (CDCl₃) spectra of crude aliquots withdrawn from the reaction system for monitoring the conversion of DGA, SA, NA, DPMA, ʟ-LA, and the formation of resultant polymers (entry 7 in Table 1).

signals of carbonyl groups of DPMA (carbons 21′ and 34′) remained unchanged. The proposed multiblock structure is further demonstrated by the following. First, the ¹H NMR and ¹³C NMR signals associated with the final multiblock structure are clearly observed (Supplementary Figs. 47 and 48). Second, the monomodal Đ is detected using SEC (Supplementary Fig. 49), and the molecular weight increases continuously with monomer

consumption, accompanied by a narrow, unimodal distribution (Đ). The Đ does not broaden until the complete consumption of ʟ-LA (Fig. 6c), and the increase of Đ value was observed during the DPMA/epoxide ROCOP step. Due to the absence of new carbonyl groups caused by transesterification side reactions during the polymerization process (Supplementary Figs. 46 and 48), we can reasonably deduce that very slow consumption of

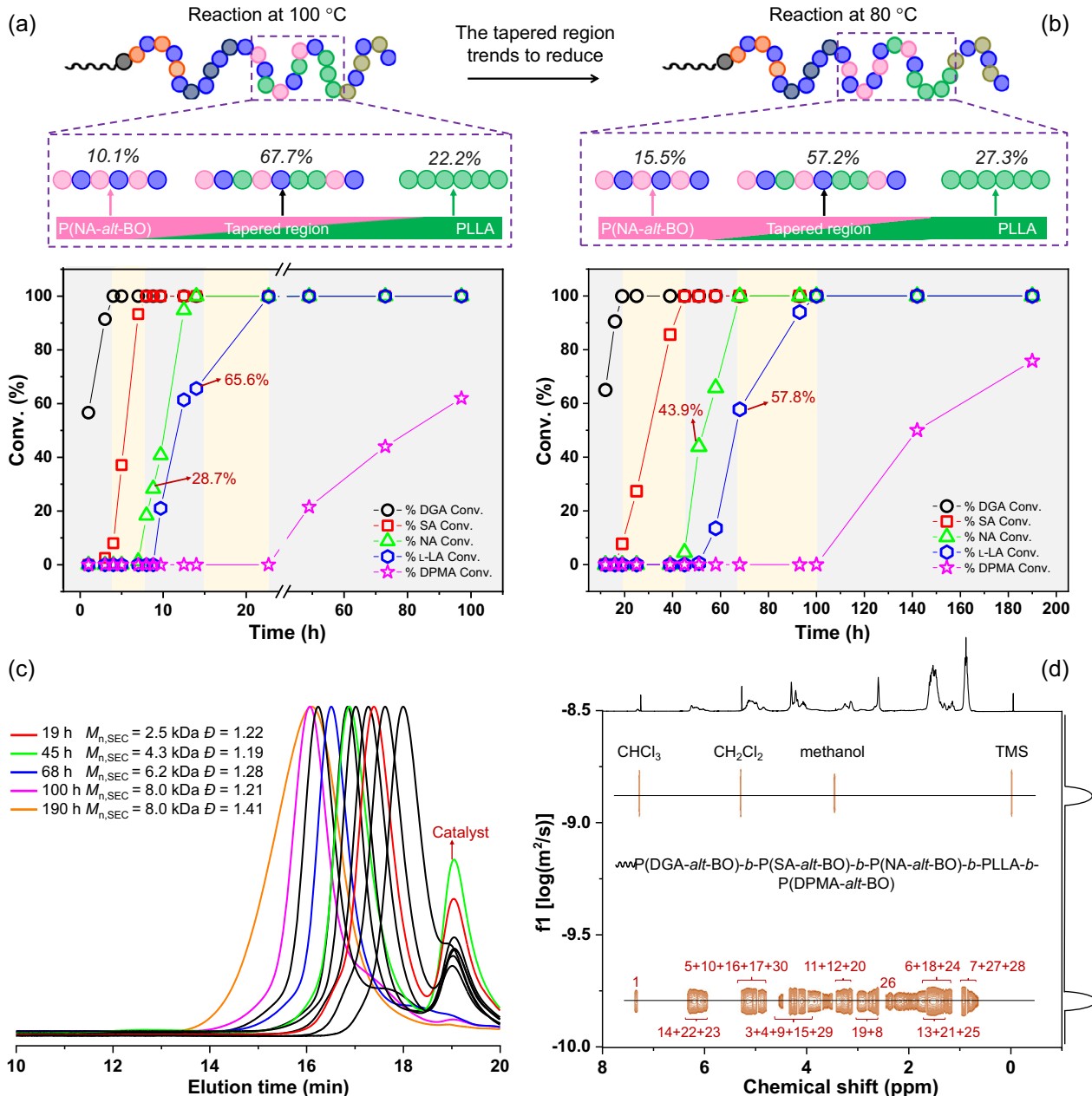

**Fig. 6 NMR and SEC analysis of the self-switchable polymerization from the mixture of DGA/SA/NA/ʟ-LA/DPMA/BO.** Plots of monomer conversion versus time as monitored by $^1$H NMR (CDCl$_3$) spectroscopy: **a** 100 °C. The tapered region includes 71.3% of NA and 65.6% of ʟ-LA, and the molar mass fraction of P(NA-*alt*-BO) segment, tapered region, and PLLA segment are 10.1%, 67.7%, and 22.2%, respectively. **b** 80 °C. The tapered region includes 56.1% of NA and 57.8% of ʟ-LA, and the molar mass fraction of P(NA-*alt*-BO) segment, tapered region, and PLLA segment are 15.5%, 57.2%, and 27.3%, respectively. **c** Evolution of SEC traces (THF) (entry 8 in Table 1). **d** DOSY (CDCl$_3$) spectrum of resultant polymers (entry 8 in Table 1).

DPMA by ROCOP of DPMA/epoxides leads to slow initiation, which makes the Đ become broad. Third, only one diffusion coefficient is observed in the DOSY of the final polymer (Fig. 6d). Although a tapered region is formed in the multiblock polymer due to the insufficient difference in the reactivities of NA and ʟ-LA, accurate sequence-defined multiblock polymers are easily obtained by the rational design of the systems, such as DGA/SA/ʟ-LA/DPMA/BO or DGA/SA/NA/DPMA/BO (Supplementary Figs. 50–54 and entry 9 in Table 1). To further increase the diversity of the multiblock polymers, polyethylene glycol (molecular weight $M_n = 2$ kDa, PEG2000) was used as a bidirectional initiator for the polymerization of DGA/SA/NA/ʟ-LA/DPMA/BO. During the reaction, appropriate control is maintained until complete insertion of ʟ-LA. The Đ broadens after full consumption of ʟ-LA where the reason has been analyzed in the DGA/SA/NA/ʟ-LA/DPMA/BO with BDM-initiator system, finally resulting in multiblock polymers consisting of up to 11 blocks (entry 10 in Table 1, Supplementary Tables 3 and 4, Supplementary Figs. 55–62).

**One-step synthesis of a core–shell-type multiblock copolymer.** Having established a switchable polymerization system, we investigated expanding this system to a functional anhydride, such as trimellitic anhydride (TA). TA favors the synthesis of core–shell-type multiblock copolymer because it has an

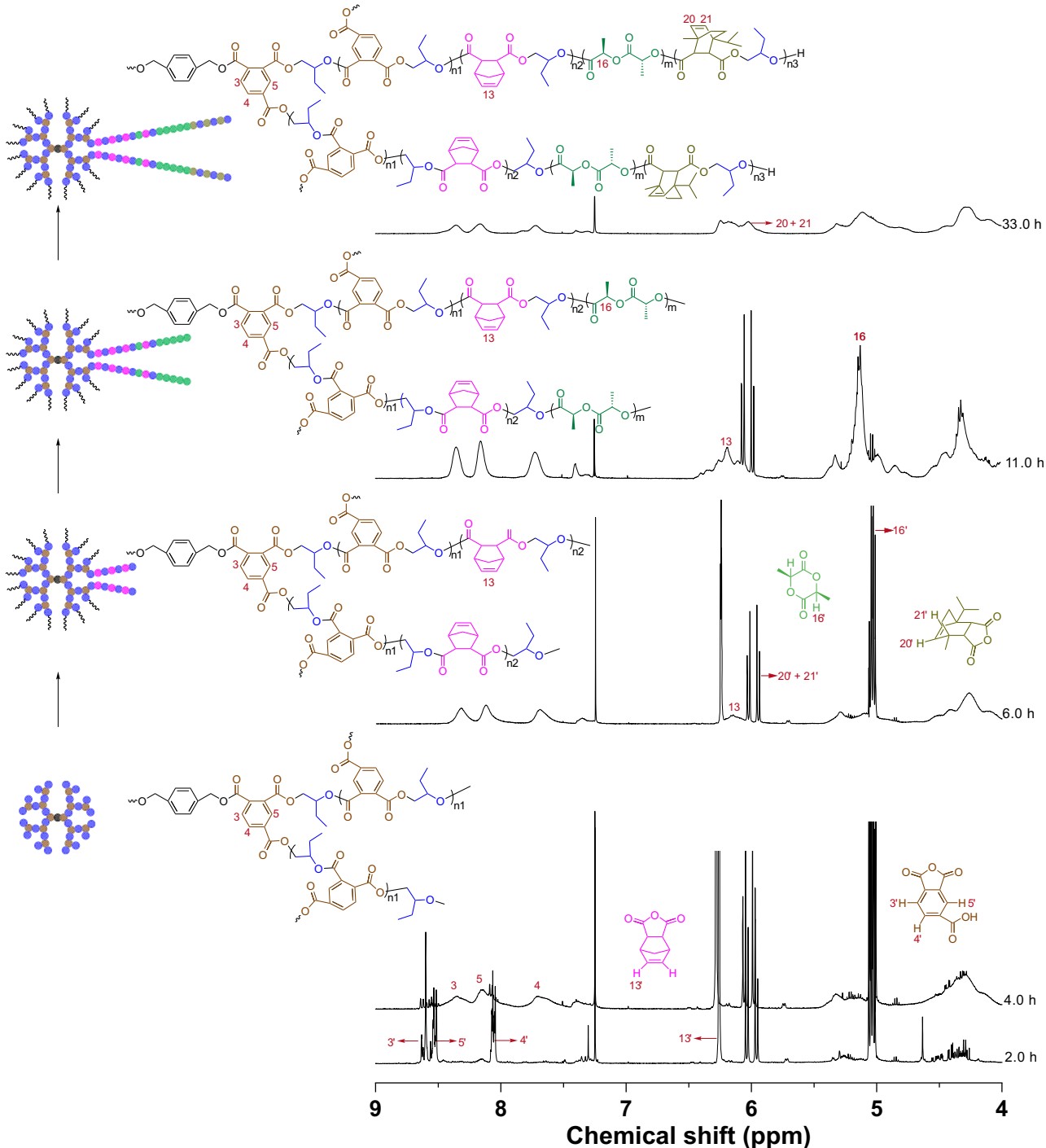

**Fig. 7 ¹H NMR (CDCl₃) spectra of the aliquots withdrawn from the polymerization of mixed TA, NA, ʟ-LA, DPMA, and BO with pivalate cesium as the catalyst and BMD as the initiator at 80 °C.** TA trimellitic anhydride.

additional carboxylic acid group that can act as a propagation site for ROCOP[35,47]. Core–shell-type multiblock copolymer has demonstrated several characteristics when compared with linear polymer, including a large population of terminal functional groups, lower melt viscosity, and better solubility[48]. Thus, it has attracted more and more attention from the scientific and engineering points of view. However, the synthesis of a core–shell-type multiblock copolymer is still limited to multistep procedure. To prepare diverse, hyperbranched architectures using the one-step procedure, we examined the reactivity ratio of TA and NA,

and the results indicate that TA is substantially more reactive than NA (Supplementary Figs. 63 and 64). Combined with the obtained reactivity trend of the comonomers, the reactivity trend is TA ≫ NA > ʟ-LA ≫ DPMA. Thus, the polymerization system of TA/NA/ʟ-LA/DPMA/BO was prepared to synthesize a core–shell-type multiblock copolymer with hyperbranched P(TA-*alt*-BO) and P(NA-*alt*-BO)-*b*-PLLA-*b*-P(DPMA-*alt*-BO) as the core and outer shells, respectively. The evolution of the complex structure was successfully monitored using NMR spectroscopy. As shown in Fig. 7, peaks at 8.64, 8.55, and 8.09 ppm representing

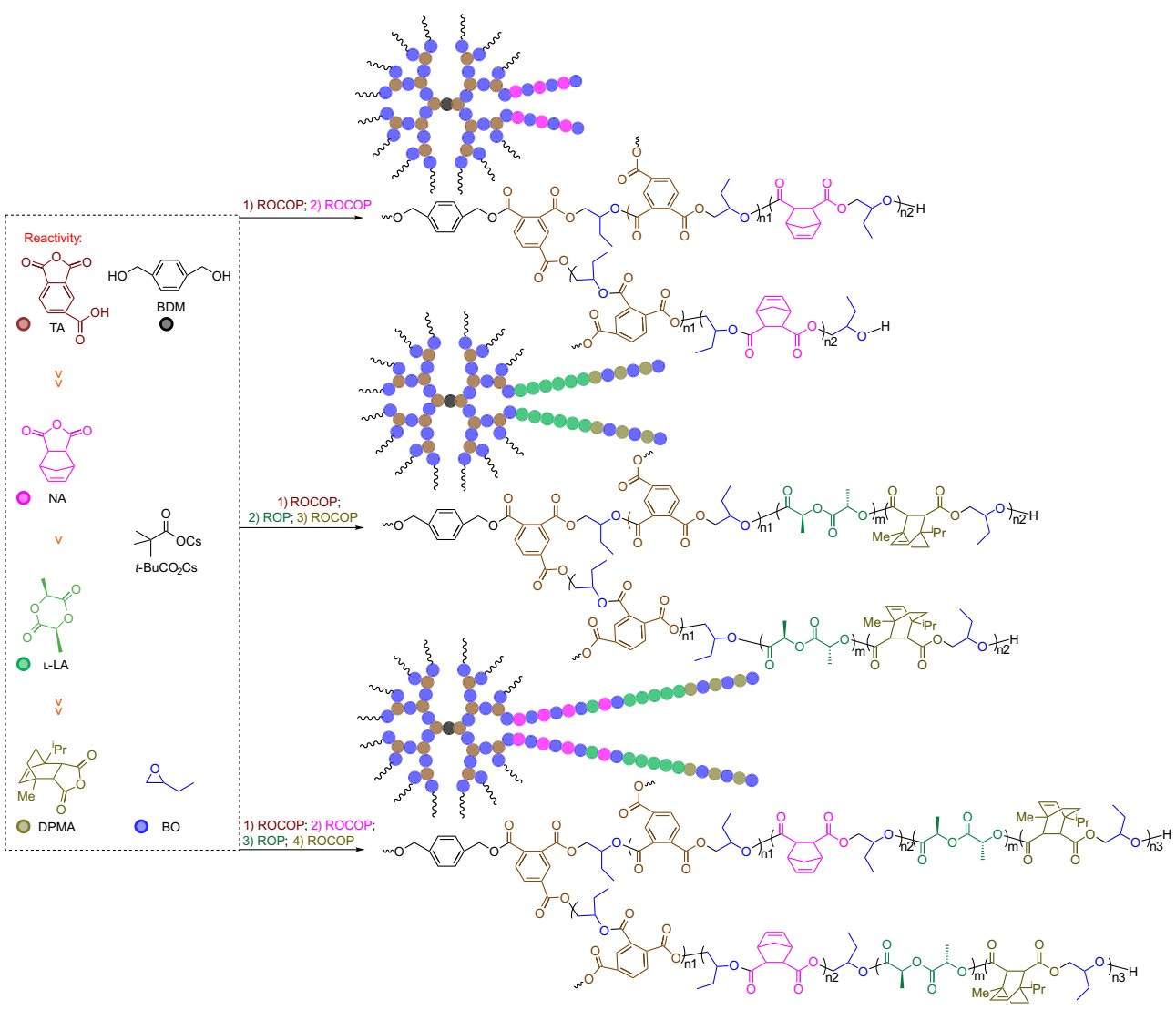

**Fig. 8 The one-step synthesis of various core–shell-type multiblock copolymers by the combination among different catalytic cycles at 80 °C.** Brown: TA. Pink: NA. Green: L-LA. Olive: DPMA. Blue: BO.

TA clearly decrease (conversion of TA reaches 93.2% at 4 h), while the three new peaks at 8.45−8.21, 8.21−8.00, and 7.79−7.52 ppm (protons 3, 4, and 5, Fig. 7) are observed, indicating formation of a hyperbranched P(TA-*alt*-BO). Concurrently, NA remains unreacted with no trace of P(NA-*alt*-BO) formation because the peak at 6.37−6.13 ppm (proton 13, Fig. 7) is not observed. The ROCOP of NA/BO commences upon 99% conversion of TA at 5.5 h, as shown by the decrease in the peak at 6.30 ppm and the observation of a new peak at 6.37−6.13 ppm (Supplementary Fig. 65). Upon 42.8% conversion of NA, the ROP of L-LA commences. Upon 99% conversion of L-LA, gradual consumption of DPMA by ROCOP of DPMA/BO occurs (Fig. 7 and Supplementary Figs. 65–69). The resultant polymer (degree of branching of ~0.68, Supplementary Fig. 68) displays a broad SEC trace and high dispersity (Đ = 1.60, Supplementary Fig. 70 and entry 11 in Table 1), which is likely owing to nonuniform branching. The diffusion-ordered NMR spectrum reveals a single diffusion coefficient for the observed signals, suggesting that only one core–shell-type multiblock copolymer exists, not a blend (Supplementary Fig. 71). Therefore, this catalytic system provides an efficient, simple, one-step procedure instead of conventional stepwise synthetic procedures for preparing core–shell-type

multiblock copolymers[49–51]. Furthermore, the structure of the outer shell, particularly the number of segments, can be flexibly regulated by rational design of the polymerization system (entries 12 and 13 in Table 1, Fig. 8, and Supplementary Figs. 72–79), offering a simple method for creating various supramolecular polymers.

In conclusion, a versatile, direct, one-step synthesis of a sequence-controlled multiblock polyester of up to 11 blocks from a six-component mixture was demonstrated. The alkali-metal carboxylate catalyst spontaneously connected five catalytic cycles, involving four cyclic anhydride/epoxide ROCOPs and an L-LA ROP. Control over the monomer-incorporation sequence based on reactivity ratio of these monomers (DGA ≫ SA ≫ NA > L-LA ≫ DPMA and TA ≫ NA > L-LA ≫ DPMA) rendered the switchable polymerization similar to ideal examples in nature, allowing the synthesis of different sequence-controlled multiblock copolymers even containing various hyperbranched architectures with the relative broad Đ (1.23−1.71). Although the obvious tapered region was formed when combining L-LA and NA because of their similar reactivity, nearly perfect multiblock polymers can be obtained by rational combination of different polymerization cycles. A notable advantage of this method was

the ability to freely manipulate the polymerization order between anhydride/epoxide ROCOP and L-LA ROP, creating a more flexible polymerization pathway. Thus, the simple and sequence-controlled polymerization yielded tailored functional materials for high-value emerging applications, such as data storage, anticounterfeiting technologies, microelectronics, and nanomedicine. However, the essential factor that determined the reactivity differences between these monomers is yet to be determined. Ongoing studies are focusing on revealing this factor and extending the applicability to a large library of structurally and functionally diverse cyclic anhydrides, epoxides, and cyclic esters.

## Methods

**General: the self-switchable polymerization protocol**. In an argon-filled glovebox, the catalyst, initiator, and monomers were added to an oven-dried reaction vessel with a magnetic stir. The reaction mixture was stirred under an argon atmosphere in an oil bath. During polymerization, a crude aliquot was time-regularly obtained from the system by a syringe in an argon flow and monitored by $^1$H NMR spectroscopy and SEC, at 30 °C using THF as the eluent and narrow molar mass polystyrene calibrants, to determine monomer conversion and molar mass. After the defined time, the polymerization was terminated by diluting the reaction mixture with dichloromethane ($CH_2Cl_2$). The reaction mixture was purified by reprecipitation from a $CH_2Cl_2$ solution into cold methanol. The purified polymers were dried under vacuum at room temperature for next analysis. A representative procedure involved adding the mixture of cesium pivalate (0.02 mmol, 1 equiv.), BDM (0.04 mmol, 2 equiv.), DGA (0.5 mmol, 25 equiv.), SA (0.5 mmol, 25 equiv.), NA (0.5 mmol, 25 equiv.), L-LA (1.5 mmol, 75 equiv.), DPMA (0.25 mmol, 12.5 equiv.), and BO (7 mmol, 350 equiv.).

## Data availability

All data are available in the main text or supplementary materials. The data that support the findings of this study are available from the corresponding authors on request.

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

## Acknowledgements

This work was financially supported by the Japan Society for the Promotion of Science Grant-in-Aid for Scientific Research (B) (Grant Number 19H02769); the Ministry of Education, Culture, Sports, Science, and Technology of Japan Grant-in-Aid for Scientific Research on Innovative Areas (Hybrid Catalysis for Enabling Molecular Synthesis on Demand; Grant Numbers 18H04639 and 20H04798); the Frontier Chemistry Center (Hokkaido University); the Photo-excitonic Project (Hokkaido University); the Creative Research Institution (CRIS, Hokkaido University); the Science and Technology Research Program of Chongqing Municipal Education Commission (Grant Numbers: KJQN201801116 and KJQN201901109); Cultivation Plan of National Natural Science Foundation of China and Social Science Foundation Project (Grant Number: 2021PYZ03); Innovation Research Group at Institutions of Higher Education in Chongqing (Grant Number: CXQT19027). X.X. acknowledges scholarship support from the China Scholarship Council (No. 201908500030).

## Author contributions

X.X., R.S, T.G., T.I., and T.S. prepared the paper and the experimental work was supervised by X.X., T.I., and T.S.

## Competing interests

The authors declare no competing interest.
