## [Peer Review File · Nature Communications]

One-step synthesis of sequence-controlled multiblock polymers with up to 11 segments from monomer mixtureREVIEWER COMMENTS

Reviewer #1 (Remarks to the Author):

This MS describes a self-switchable copolymerization activated by simple alkali metal carboxylate catalysts that spontaneously and smartly switch between five catalytic cycles, involving four ROACs of anhydrides/epoxides and one ROP of cyclic esters resulting in a one-step synthetic pathway for preparing linear and non-linear multiblock polymers.

In my opinion, the results obtained are certainly interesting but not accurate enough or very important for publication in Nature Communications. The most important characteristics are missing (exact molecular weight, reactivity ratios, etc.).

I suggest the authors consider the following comments before submitting the MS to another journal.

1) NMR was used to monitor the entire polymerization process and to confirm all multiblock structures, so the M_n values determined by ^1H NMR should be given in Table 1 instead of the $M_n(\text{SEC})$ based on PS standards. It is wrong to compare the calculated M_n with the determined by SEC.

2) Some multiblock polymers containing PLLA segments have $\mathcal{D} > 1.5$. For example, in Figure 5d, when the polymerization time reaches 190 hours, the SEC trace is very broad (see also Table 1). Therefore, the authors should comment on it (presence of transesterification and back-biting reactions).

3) The synthesis of sequence-controlled multiblock polymers is based on the reactivity ratios of the different monomers (anhydrides, epoxides, and LLA). The authors should determine and report the different reactivity ratios.

4) The presence of random/tapered segments between the two blocks should be quantified by NMR in all cases.

5) The characteristic peaks of polymers in the ^1H NMR spectra should be zoomed in, especially in the switch areas (Figure 5).

6) The ^{13}C NMR spectra should be used to observe the carbonyl groups and trace the random/tapered sequences.

Reviewer #2 (Remarks to the Author):

Smart one-step synthesis of sequence-controlled multiblock polymers consisting of up to 11 segments from monomer mixture

Xiaochao Xia, Ryota Suzuki, Tianle Gao, Takuya Isono, Toshifumi Satoh

The authors report a Cs(pivalate) catalyst for the polymerisation of 4 different anhydride monomers, different epoxides and lactide to make complex block polymers, some featuring upto 11 different blocks, in one pot. In general this is nicely conducted work and there's no doubt that interesting catalysis is reported and exciting new polymers have been produced. I also really like the concept of taking advantage of different reactivity ratios of anhydrides, coupled with switch catalysis to make highly selective block polymers from really complex monomer mixtures.

Although there are some detractions to the system (e.g. specific anhydride combinations do need to be selected) and it could be argued that the concepts are quite obvious based on the prior rules for switch catalysis, I feel this work does meet the high bar at Nature communications for the following reasons. 1) The ability to selectively enchain mixtures of 5 and 6 different monomers using different mechanisms is really impressive and an advance in this field, 2) The catalyst is really straightforward to apply and this will allow polymer chemists to take advantage, 3) the use of a monomer with an initiating group attached is a neat way to build more complex structures and is novel. Thus, I feel the work warrants publication at JACS. There are some aspects which need to be tidied and improved prior to publication.

General comments

1) Figures are difficult to interpret (overly complex and cluttered) and the authors should really consider using the same colour schemes as others in the field are using (red = CO₂, blue = epoxide, green = lactone, pink = anhydride). This, with the same nomenclature (ROCOP in place of ROAC and switchable polymerization consistently instead of smart polymerization which occurs at points).

2) Polymerizations are not well controlled, $\mathcal{D} = 1.3 - 1.7$. This means there may be some

transesterification side-reactions responsible for block scrambling? The authors should certainly prove this is not happening (¹³C NMR can be useful here). perhaps some studies into ROP and ROCOP reactions can establish why the dispersity is high – e.g. initiation problems?

3) Overall the polyester molar masses are low, <8 kDa, with Entry 4 at 2 kDa. What are the block DP for each segment and what is the confidence that all these blocks are present in the structure? Perhaps this could be addressed by stating the limits of catalyst loading and, therefore, molar mass?

4) It's a bit difficult to see the motivation for making a star/dendritic copolymer and some proof of concept rheology or application testing would be really beneficial to illustrate why the community would adopt this approach.

Reviewer #3 (Remarks to the Author):

When multiple number of monomers differing in reactivity co-exist, the more reactive one gets consumed over the other(s). This is rather general especially in ionic polymerizations.

Here in this manuscript, the authors selected many monomers to accomplish a world record of the number of blocks. Although the concept itself is not new, I think it is still remarkable that they could come to this fine selection of monomers.

While I found the well-defined polymer structure is fascinating, it is unfortunate that the expressions used in this manuscript is somewhat misleading in many cases, for example,

1. "The catalyst smartly switches" in Abstract, line 13. I don't think any catalyst is "smart" or "dull". It simply selects the monomers in the order of activation energy.

2. Also, the word "switch" is not suitable. Simply. once one monomer is consumed, the next monomer starts to react.

3. The introduction is quite misleading. The polymer obtained in this work still varies in the unit numbers of each block, which is contrastive to the perfectly controlled sequence of biopolymers such as nucleic acids or proteins. Given that the method presented here inherently never reach the level of control of biopolymers, the introduction should be completely rewritten without the unnecessary comparisons.

In conclusion, the authors here report an advanced, more complicated sequence of block copolymers in a remarkably sophisticated manner using previously reported strategy in ref 26.

Additional minor points:

Time course profile of SEC peak should be added in the supporting information (Not only the final product).

In the initiating group, RO- oxygen is missing in Figure 1.

We appreciate the comments from the editor and the reviewers. The following is our response to the comments.

Reviewer: 1

General comments:

This MS describes a self-switchable copolymerization activated by simple alkali metal carboxylate catalysts that spontaneously and smartly switch between five catalytic cycles, involving four ROACs of anhydrides/epoxides and one ROP of cyclic esters resulting in a one-step synthetic pathway for preparing linear and non-linear multiblock polymers.

In my opinion, the results obtained are certainly interesting but not accurate enough or very important for publication in Nature Communications. The most important characteristics are missing (exact molecular weight, reactivity ratios, etc.).

I suggest the authors consider the following comments before submitting the MS to another journal.

Response: Thank you very much for your helpful comments! It is really helpful for us to improve our work. We have made major revisions according to your comments.

Comment 1:

NMR was used to monitor the entire polymerization process and to confirm all multiblock structures, so the M_n values determined by ^1H NMR should be given in Table 1 instead of the $M_n(\text{SEC})$ based on PS standards. It is wrong to compare the calculated M_n with the determined by SEC.

Response: Thank you very much for your careful checking and kind suggestions. We agree with your comments, and the $M_{n, \text{NMR}}$ values have been added in Table 1. Because the NMR based molecular weights of the multi-block polymer from DGA/SA/NA/L-LA/DPMABO with PEG as initiator is not very precise, $M_{n, \text{MALS}}$ was obtained using SEC with multi-angle light scattering detector (SEC-MALS) and the value is 18.7 kDa, which is close to $M_{n, \text{th}}$ (16.5 kDa). The revisions were marked in red color on page 3 in the revised manuscript.

Before revision

Page 3. **Table 1.** The self-switchable polymerizations catalyzed by cesium pivalate.^a

entry	monomers	[anhydride] _a /	Temp.	Time	conv. ^b [%]	$M_{n,th}$	$M_{n,SEC}$	\bar{D}^d
		[L-LA] _a /[epoxide] _a /[BDM] _a /[cat.] _b						
1	DGA/NA/EGE	25/25/150/2/1	100	6	DGA > 99, NA = 92	5,9	5,8	1,37
2	DGA/SA/NA/DPMA/BO	25/25/25/12,5/250/2/1	100	73	DGA > 99, SA > 99, NA > 99, DPMA = 86	9,2	7,4	1,30
3	DGA/SA/NA/EGE	25/25/25/250/2/1	100	7	DGA > 99, SA > 99, NA = 71	7,7	6,9	1,28
4	L-LA/DPMA/EGE	50/25/250/2/1	100	23	L-LA > 99, DPMA = 80	6,7	2,0	1,51
5	DPMA/BO	25/125/2/1	80	41	DPMA > 99	3,9	2,1	1,16
6	L-LA/DPMA/BO	50/25/250/2/1	80	41	L-LA > 99, DPMA = 45	6,5	4,2	1,35
7	DGA/SA/NA/L-LA/DPMA/BO	25/25/25/75/12,5/350/2/1	100	97	DGA > 99, SA > 99, NA > 99, L-LA > 99, DPMA = 86	14,6	6,6	1,51
8	DGA/SA/NA/L-LA/DPMA/BO	25/25/25/75/12,5/350/2/1	80	190	DGA > 99, SA > 99, NA > 99, L-LA > 99, DPMA = 76	14,4	8,0	1,41
9	DGA/SA/L-LA /DPMA/BO	25/25/75/12,5/350/2/1	100	97	DGA > 99, SA > 99, L-LA > 99, DPMA = 41	10,8	6,8	1,23
^e 10	DGA/SA/NA/L-LA/DPMA/BO	25/25/25/75/12,5/350/2/1	80	66	DGA > 99, SA > 99, NA > 99, L-LA > 99, DPMA = 91	16,5	7,8	1,46
11	TA/NA/L-LA/DPMA/BO	25/25/50/12,5/350/2/1	80	24	TA > 99, NA > 99, L-LA > 99, DPMA = 54,5	11,0	7,7	1,60
12	TA/NA /BO	25/25/150/2/1	80	9	TA > 99, NA = 60,5	5,2	4,3	1,61
13	TA/L-LA/DPMA/BO	25/50/12,5/350/2/1	80	25	TA > 99, L-LA > 99, DPMA = 50	8,0	5,1	1,71

After revision

Table 1. The self-switchable polymerizations catalyzed by cesium pivalate.^a

entry	monomers	[anhydride] _o /	Temp. [°C]	Time (h)	conv. ^b [%] (anhydride or L-LA)	M _{n,th.} [kDa]	M _{n,NMR.} ^b [kDa]	M _{n,SEC} ^d [kDa]	D ^d
		[L-LA] _o /[epoxide] _o /[BDM] _o /[cat.] _o							
1	DGA/NA/EGE	25/25/150/2/1	100	6	DGA > 99, NA = 92	5,9	6,8	5,8	1,37
2	DGA/SA/NA/DPMA/BO	25/25/25/12,5/250/2/1	100	73	DGA > 99, SA > 99, NA > 99, DPMA = 86	9,2	9,8	7,4	1,30
3	DGA/SA/NA/EGE	25/25/25/250/2/1	100	7	DGA > 99, SA > 99, NA = 71	7,7	7,2	6,9	1,28
4	DPMA/L-LA/EGE	25/50/250/2/1	100	23	L-LA > 99, DPMA = 80	6,7	7,2	2,0	1,51
5	DPMA/BO	25/125/2/1	80	41	DPMA > 99	3,9	n.d.	2,1	1,16
6	DPMA/L-LA/BO	25/50/250/2/1	80	41	L-LA > 99, DPMA = 45	5,3	6,4	4,2	1,35
7	DGA/SA/NA/L-LA/DPMA/BO	25/25/25/75/12,5/350/2/1	100	97	DGA > 99, SA > 99, NA > 99, L-LA > 99, DPMA = 86	14,6	14,9	6,6	1,51
8	DGA/SA/NA/L-LA/DPMA/BO	25/25/25/75/12,5/350/2/1	80	190	DGA > 99, SA > 99, NA > 99, L-LA > 99, DPMA = 76	14,4	15,4	8,0	1,41
9	DGA/SA/L-LA /DPMA/BO	25/25/75/12,5/350/2/1	100	97	DGA > 99, SA > 99, L-LA > 99, DPMA = 41	10,8	11,1	6,8	1,23
°10	DGA/SA/NA/L-LA/DPMA/BO	25/25/25/75/12,5/350/2/1	80	66	DGA > 99, SA > 99, NA > 99, L-LA > 99, DPMA = 91	16,5	15,5	7,8, 18,7 ^f	1,46
11	TA/NA/L-LA/DPMA/BO	25/25/50/12,5/350/2/1	80	24	TA > 99, NA > 99, L-LA > 99, DPMA = 54,5	11,9	12,2	7,7	1,60
12	TA/NA /BO	25/25/150/2/1	80	10	TA > 99, NA = 60,5	6,0	5,8	4,3	1,61
13	TA/L-LA/DPMA/BO	25/50/12,5/350/2/1	80	25	TA > 99, L-LA > 99, DPMA = 50	8,9	9,3	5,1	1,71

^a Polymerization conditions: Ar atmosphere. ^b Determined by ¹H NMR analysis of the obtained polymer in CDCl₃. ^c Theoretical *M*_n values. ^d Determined by the SEC analysis of the obtained polymer in THF with a PSt standard. ^e PEG2000 is used as a bidirectional initiator. ^f Determined by SEC with multi-angle light scattering detector (SEC-MALS).

Comment 2:

Some multiblock polymers containing PLLA segments have $\bar{D} > 1.5$. For example, in Figure 5d, when the polymerization time reaches 190 hours, the SEC trace is very broad (see also Table 1). Therefore, the authors should comment on it (presence of transesterification and back-biting reactions).

Response: Thank you very much for your careful checking and kind suggestions. We agree with your comments, and we have commented why some multiblock polymers containing PLLA segments have $\bar{D} > 1.5$. The polyester from Entry 4 showed a broad \bar{D} (1.51), which could be attributed to the transesterification or back-biting reactions at 100 °C, but the side reaction can be suppressed by lowering the temperature from 100 °C to 80 °C (entry 6 in Table 1, $\bar{D} = 1.35$, $M_{n,SEC} = 4.2$ kDa, Figure S37). We have also tried to use ^{13}C NMR (including ^{13}C NMR spectra of the final product and crude aliquots withdrawn from the reaction system) to analyze whether transesterification side-reactions occurs or not. Since the final ^{13}C NMR spectra (Figure S48) includes many carbons, it was very difficult to identify the transesterification products by assigning each signal. Therefore, we used ^{13}C NMR spectra to monitor the polymerization processing, and we did not observe new carbonyl groups caused by transesterification side-reactions during the polymerization process (Figures S46 and S48). The result indicated that transesterification side-reactions should not occur. The SEC trace showed that dispersity remains narrow, and it became large only when DPMA started to consume. This can be attributed to slow initiation caused by the very slow consumption of DPMA. Relevant discussion was added in the revised manuscript. The revisions were marked in red color.

Figure S46. The ¹³C NMR (CDCl₃) spectra of crude aliquots withdrawn from the reaction system for monitoring the conversion of DGA, SA, NA, DPMA, L-LA, and the formation of resultant polymers (entry 8 in Table 1).

Addition

Page 5 in the revised manuscript. This is due to transesterification or back-biting reactions could occur during polymerization for DPMA/L-LA/EGE system. The polymerization from the mixture of DPMA/L-LA/BO was carried out at 80 °C, and the resultant triblock polymer displayed the better controllability ($\bar{D} = 1.35$, entry 6 in table, Figures S37–S39) relative to DPMA/L-LA/EGE system. This result indicated that side reactions such as transesterification or back-biting reactions can be suppressed by lowering reaction temperature.

Page 6 the revised manuscript. The \bar{D} do not broaden until the complete consumption of L-LA (Figure 5d), and the increase of \bar{D} value was observed during the DPMA/epoxide ROCOP step. Due to the absence of new carbonyl groups caused by transesterification side-reactions during the polymerization process (Figures S46 and S48), we can reasonably deduce that very slow consumption of DPMA by ROCOP of

DPMA/epoxides leads to slow initiation, which makes the \mathcal{D} become broad.

Comment 3:

The synthesis of sequence-controlled multiblock polymers is based on the reactivity ratios of the different monomers (anhydrides, epoxides, and LLA). The authors should determine and report the different reactivity ratios.

Response: Thank you very much for your careful checking and kind suggestions. The synthesis of sequence-controlled multiblock polymers is based on the reactivity ratios of anhydrides coupled with switch catalysis. In our previous work,^{R1} we showed that the carbonyl group of the anhydride is activated by a cesium cation. Owing to their extremely high electrophilicity, the activated anhydride is prone to nucleophilic attack from the cesium pivalate-activated hydroxyl group rather than the epoxide or cyclic ester. The resulting carboxylate species are unable to react with anhydride and lactide (Figure R1), and can only react with the cesium cation-activated epoxide. Although hydroxyl species came from ring-opening of epoxide can react with both anhydride and epoxide, the reactivity of anhydride is much higher than that of epoxide so that hydroxyl species only react with anhydrides to form perfect alternating chemical structure (Figure R2). Therefore, the reactivity ratio of anhydride and epoxide is not the only factor that leads to the ring-opening alternating copolymerization (ROCOP) of anhydride with epoxide. In our previous work, we performed a polymerization using cesium pivalate, PPA, ethyl glycidyl ether (EGE), and L-LA with a molar ratio of 1/2/250/100 at 100 °C. After 35 min, PLLA with a narrow \mathcal{D} of 1.20 was obtained with ~93.5% L-LA conversion and no ROP of EGE occurred. When the reaction was prolonged to 19 h, conversion of L-LA reached 100%, while EGE did not self-propagate (Figure R3). Therefore, reactivity ratio of L-LA and epoxide is unable to be obtained in the cesium pivalate-catalyzed polymerization system. Polymerization order between ROCOP of anhydride with epoxide and ring-opening polymerization (ROP) of L-LA depends on the reactivity ratio of anhydride and L-LA. Polymerization order between ROCOP of two different anhydrides with epoxide depends on the reactivity ratio of the two anhydrides. Reactivity ratio of different anhydrides and reactivity ratio of anhydride and L-LA can be described by the nonterminal model of copolymerization kinetics.^{R2,R3} Here, p_A and p_B are the respective conversions of A and B monomer with $p_A = 1 - (A(t)/A_0)$.

$$p_{AB}(p_A) = 1 - n_A(1 - p_A) - (1 - n_A)(1 - p_A)^{r_B} \quad (1)$$

$$p_{AB}(p_B) = 1 - (1 - n_A)(1 - p_B) - n_A(1 - p_B)^{r_A} \quad (2)$$

The nonterminal model successfully yielded reactivity ratios, which were added in the revised Supporting Information and relevant discussion were also added in the revised manuscript. The revisions were marked in red color.

Figure R1. Proposed mechanism of the self-switchable polymerization from a mixture of anhydrides, epoxides, and cyclic esters with cesium pivalate as the catalyst and alcohol as the initiator.

Figure R2. (a) MALDI TOF-MS spectrum of the P(PA-*alt*-EGE) from entry 1 in Table S2, (b) expanded spectrum in the m/z 3610–3950 range, and (c) expected structures and theoretical molecular weights of P(PA-*alt*-EGE).

Figure R3. (b) ^1H NMR spectrum of the product obtained from a mixture of PPA, $t\text{-BuCO}_2\text{Cs}$, EGE, and L-LA heated at $100\text{ }^\circ\text{C}$ for 35 min and 19 h with the feed ratio of $[\text{t-BuCO}_2\text{Cs}]/[\text{PPA}]_0/[\text{EGE}]_0/[\text{LA}]_0$ being 1/2/250/100.

R1. Xia, X. C., Ryota, S., Takojima, K., Jiang, D. H., Isono, T., Satoh, T., Smart access to sequentially and architecturally controlled block polymers via a simple catalytic polymerization system. *ACS Catalysis*, 11, 5999-6009 (2021).

R2. Beckingham, B. S., Sanoja, G. E., & Lynd, N. A. Simple and accurate determination of reactivity ratios using a nonterminal model of chain copolymerization. *Macromolecules*, 2015, 48(19), 6922-6930.

R3. Chwatko, M., & Lynd, N. A. Statistical copolymerization of epoxides and lactones to high molecular weight. *Macromolecules*, 2017, 50(7), 2714-2723.

Addition

Page 4 in the revised manuscript. The nonterminal model yielded reactivity ratios of $r_{\text{DGA}} = 827.46 \pm 50.30$ and $r_{\text{NA}} = 0.0036 \pm 0.0006$ (Figure S9). On the basis of these reactivity ratios, we also concluded that the resultant polymers were most consistent with nearly perfect triblock copolymer.

Page 4 in the revised manuscript. Based on the initial findings, a series of experiments were conducted to evaluate the reactivity ratio of the anhydrides (Figures S8 and S9, Figures S11–S14), and the reactivity trend was determined as: $\text{DGA} \gg \text{SA} \gg \text{NA} \gg \text{DPMA}$ (Figure S15).

Page 5 in the revised manuscript. The nonterminal model yielded reactivity ratios (Figure S33, $r_{\text{L-LA}} = 831.91 \pm 49.22$ and $r_{\text{DPMA}} = 0.0069 \pm 0.0002$), which confirmed the perfect triblock copolymer formation.

Reactivity ratio of DGA and NA

Since the nonterminal copolymerization kinetics are common in coordination–insertion, ionic, and pseudo ionic type polymerization mechanisms, we employed the simple model for compositional drift reported by Beckingham et al. (BSL) to determine the reactivity ratio.^{1,2} Here, equations 1 and 2 enable a simple and accurate determination of reactivity ratios at all conversions for copolymerizations that can be described by the nonterminal model of copolymerization kinetics. Here, p_A and p_B are the respective conversions of A and B monomer with $p_A = 1 - (A(t)/A_0)$.

$$p_{AB}(p_A) = 1 - n_A(1 - p_A) - (1 - n_A)(1 - p_A)^{r_B} \quad (1)$$

$$p_{AB}(p_B) = 1 - (1 - n_A)(1 - p_B) - n_A(1 - p_B)^{r_A} \quad (2)$$

1. Beckingham, B. S., Sanoja, G. E., & Lynd, N. A. Simple and accurate determination of reactivity ratios using a nonterminal model of chain copolymerization. *Macromolecules*, 2015, 48(19), 6922-6930.

2. Chwatko, M., & Lynd, N. A. Statistical copolymerization of epoxides and lactones to high molecular weight. *Macromolecules*, 2017, 50(7), 2714-2723.

Figure S9. Total polymerization conversion plotted against monomer conversion and the data were obtained from time-resolved ¹H NMR spectra of a cesium pivalate-catalyzed copolymerization of DGA/NA/EGE with a molar ratio of 25/25/150 at 100 °C. Solid black and red lines represent fits to the experimental data using the nonterminal model, eqs 1 and 2.

Reactivity ratio of different anhydrides

Figure S14. Total polymerization conversion plotted against monomer conversion and the data were obtained from time-resolved ^1H NMR spectra of a cesium pivalate-catalyzed copolymerization of DGA/SA/BO, SA/NA/BO, and NA/DPMA/BO with a molar ratio of 25/25/150 at 80 °C, respectively. Solid black and red lines represent fits to the experimental data using the nonterminal model, eqs 1 and 2.

Figure S33. Total polymerization conversion plotted against monomer conversion the data were obtained from time-resolved ^1H NMR spectra of a cesium pivalate-catalyzed copolymerization of DPMA/L-LA/EGE with a molar ratio of 25/50/250 at 100 °C. Solid black and red lines represent fits to the experimental data using the nonterminal model, eqs 1 and 2.

Figure S44. Total polymerization conversion plotted against monomer conversion. Solid black and red lines represent fits to the experimental data using the nonterminal model, eqs 1 and 2. (a) copolymerization of NA/EGE/L-LA with a molar ratio of 25/50/250 at 100 °C; (b) copolymerization of NA/EGE/L-LA with a molar ratio of 25/50/250 at 50 °C.

Figure S64. Total polymerization conversion plotted against monomer conversion the data were obtained from time-resolved ^1H NMR spectra of a cesium pivalate-catalyzed copolymerization of TA/NA/BO with a molar ratio of 25/25/150 at 80 °C. Solid black and red lines represent fits to the experimental data using the nonterminal model, eqs 1 and 2.

Comment 4:

The presence of random/tapered segments between the two blocks should be quantified by NMR in all cases.

Response: Thank you very much for your careful checking and kind suggestions. According to your comments, the random segments between the blocks have been quantified by ^1H NMR. The revision was marked in red color in the revised manuscript and revised Supporting Information.

After revision

Page 7 in the revised manuscript.

Figure 6. Plots of monomer conversion versus time as monitored by ^1H NMR (CDCl_3) spectroscopy: (a) 100 °C; The tapered region includes 71.3% of NA and 65.6% of L-LA, and the molar mass fraction of P(NA-*b*-BO) segment, tapered region, and PLLA segment are 10.1%, 67.7%, and 22.2%, respectively; (b) 80 °C; The tapered region includes 56.1% of NA and 57.8% of L-LA, and the molar mass fraction of P(NA-*b*-BO) segment, tapered region, and PLLA segment are 15.5%, 57.2%, and 27.3%, respectively. (c) Evolution of SEC traces (THF) (entry 8 in Table 1). (d) DOSY (CDCl_3) spectrum of resultant polymers (entry 8 in Table 1).

Figure S55. The ^1H NMR (CDCl_3) spectra of crude aliquots withdrawn from the reaction system for monitoring the conversion of DGA, SA, NA, DPMA, L-LA, and the formation of resultant polymers (entry 10 in Table 1).

Figure S57. The plots of monomer conversion versus time. The polymerization of DGA, SA, NA, DPMA, L-LA with BO was performed at 80 °C (entry 10 in Table 1). The tapered region includes 51.5% of NA and 57.3% of L-LA, and the molar mass fraction of P(NA-*b*-BO) segment, tapered region, and PLLA segment are 17.1%, 55.3%, and 27.6%, respectively.

Figure S65. The ¹H NMR (CDCl₃) spectra of crude aliquots withdrawn from the reaction system for monitoring the conversion of TA, NA, DPMA, L-LA, and the formation of resultant polymers (entry 11 in Table 1).

Figure S67. The plots of monomer conversion versus time. The polymerization of TA, NA, DPMA, L-LA with BO was performed at 80 °C (entry 11 in Table 1). The tapered region includes 57.2% of NA and 61.1% of L-LA, and the molar mass fraction of P(NA-*b*-BO) segment, tapered region, and PLLA segment are 19.3%, 59.3%, and 21.4%, respectively.

Comment 5:

The characteristic peaks of polymers in the ¹H NMR spectra should be zoomed in, especially in the switch areas (Figure 5).

Response: Thank you very much for your careful checking and kind suggestions. According to your comments, the ¹H NMR spectra in the switch areas have been zoomed in for all case. The revision was marked in red color in the revised Supporting Information.

After revision

Page 10 in the Supporting Information.

Figure S1. The ¹H NMR (CDCl₃) spectra of crude aliquots withdrawn from the reaction system for monitoring the conversion of DGD and NA and the formation of resultant triblock polymers (entry 1 in Table 1).

Figure S16. The ^1H NMR (CDCl_3) spectra of crude aliquots withdrawn from the reaction system for monitoring the conversion of DGA, SA, NA, DPMA, and the formation of resultant heptablock polymers (entry 2 in Table 1).

Figure S22. The ¹H NMR (CDCl₃) spectra of crude aliquots withdrawn from the reaction system for monitoring the conversion of DGA, SA, NA, and the formation of resultant polymers (entry 3 in Table 1).

Figure S40. The ¹H NMR (CDCl₃) spectra of crude aliquots withdrawn from the reaction system for monitoring the conversion of DGA, SA, NA, DPMA, L-LA, and the formation of resultant polymers. Red line (entry 7 in Table 1) and black line (entry 8 in Table 1)

Figure S41. The ^1H NMR (CDCl_3) spectra of crude aliquots withdrawn from the reaction system for monitoring the conversion of DGA, SA, NA, DPMA, L-LA, and the formation of resultant polymers. Red line (entry 7 in Table 1) and black line (entry 8 in Table 1)

Figure S42. The ^1H NMR (CDCl_3) spectra of crude aliquots withdrawn from the reaction system for monitoring the conversion of DGA, SA, NA, DPMA, L-LA, and the formation of resultant polymers. Red line (entry 7 in Table 1) and black line (entry 8 in Table 1)

Page 42 in the Supporting Information.

Figure S45. The ^1H NMR (CDCl_3) spectra of crude aliquots withdrawn from the reaction system for monitoring the conversion of DGA, SA, NA, DPMA, L-LA, and the formation of resultant polymers. Red line (entry 7 in Table 1) and black line (entry 8 in Table 1)

Figure S50. The ^1H NMR (CDCl_3) spectra of crude aliquots withdrawn from the reaction system for monitoring the conversion of DGA, SA, DPMA, L-LA, and the formation of resultant polymers (entry 9 in Table 1).

Figure S55. The ^1H NMR (CDCl_3) spectra of crude aliquots withdrawn from the reaction system for monitoring the conversion of DGA, SA, NA, DPMA, L-LA, and the formation of resultant polymers (entry 10 in Table 1).

Figure S72. The ¹H NMR (CDCl₃) spectra of crude aliquots withdrawn from the reaction system for monitoring the conversion of TA, NA, and the formation of resultant polymers (entry 12 in Table 1).

Figure S76. The ^1H NMR (CDCl_3) spectra of crude aliquots withdrawn from the reaction system for monitoring the conversion of TA, L-LA, DPMA and the formation of resultant polymers (entry 13 in Table 1).

Comment 6:

The ^{13}C NMR spectra should be used to observe the carbonyl groups and trace the random/tapered sequences.

Response: Thank you very much for your careful checking and kind suggestions. According to your comments, the ^{13}C NMR spectra has been used to observe the carbonyl groups and trace the random/tapered sequences for the important polymerization system including TA/NA/L-LA/DPMA/BO and DGA/SA/NA/L-LA/DPMA/BO initiated by BDM and PEG, respectively. The data have been added in the revised Supporting Information and marked in red color. The relevant discussion was added in the revised manuscript. The revisions were marked in red color.

Added

Page 43 in the revised Supporting Information.

Figure S46. The ^{13}C NMR (CDCl₃) spectra of crude aliquots withdrawn from the reaction system for monitoring the conversion of DGA, SA, NA, DPMA, L-LA, and the formation of resultant polymers (entry 8 in Table 1).

Figure S56. The ^{13}C NMR (CDCl₃) spectra of crude aliquots withdrawn from the reaction system for monitoring the conversion of DGA, SA, NA, DPMA, L-LA, and the formation of resultant polymers (entry 10 in Table 1).

Figure S66. The ^{13}C NMR (CDCl_3) spectra of crude aliquots withdrawn from the reaction system for monitoring the conversion of TA, NA, DPMA, L-LA, and the formation of resultant polymers (entry 11 in Table 1).

Addition

Page 6 in the revised manuscript.

The formation process of the multiblock polymer was also confirmed by the evolution of the ^{13}C NMR spectrum (Figure S46). In particular, the ^{13}C NMR signals associated with carbonyl groups of DGA (carbon 1'), SA (carbon 7'), and NA (carbon 11') disappeared one by one, while the ^{13}C NMR signals associated with carbonyl groups of P(DGA-*b*-BO) (carbon 1), P(SA-*b*-BO) (carbon 7), and P(NA-*b*-BO) (carbon 11) segments appeared in succession. After 43.9% conversion of NA, the ^{13}C NMR signals associated with carbonyl groups of L-LA (carbon 18') also commenced decreasing, accompanied by the appearance of the ^{13}C NMR signals associated with carbonyl groups of the PLLA segment (carbon 18). During the process, the signals of carbonyl groups of DPMA (carbons 21' and 34') remained unchanged.

Reviewer: 2

General comments:

The authors report a Cs(pivalate) catalyst for the polymerisation of 4 different anhydride monomers, different epoxides and lactide to make complex block polymers, some featuring upto 11 different blocks, in one pot. In general this is nicely conducted work and there's no doubt that interesting catalysis is reported and exciting new polymers have been produced. I also really like the concept of taking advantage of different reactivity ratios of anhydrides, coupled with switch catalysis to make highly selective block polymers from really complex monomer mixtures. Although there are some detractions to the system (e.g. specific anhydride combinations do need to be selected) and it could be argued that the concepts are quite obvious based on the prior rules for switch catalysis, I feel this work does meet the high bar at Nature communications for the following reasons. 1) The ability to selectively enchain mixtures of 5 and 6 different monomers using different mechanisms is really impressive and an advance in this field, 2) The catalyst is really straightforward to apply and this will allow polymer chemists to take advantage, 3) the use of a monomer with an initiating group attached is a neat way to build more complex structures and is novel. Thus, I feel the work warrants publication at JACS. There are some aspects which need to be tidied and improved prior to publication.

Thank you very much for your helpful comments! It is really helpful for us to improve our work. We have made major revisions according to your comments.

Comment 1:

Figures are difficult to interpret (overly complex and cluttered) and the authors should really consider using the same color schemes as others in the field are using (red = CO₂, blue = epoxide, green = lactone, pink = anhydride). This, with the same nomenclature (ROCOP in place of ROAC and switchable polymerization consistently instead of smart polymerization which occurs at points).

Response: Thank you very much for your careful checking and kind suggestions. According to your comments, the color schemes same to that in the field have been used in the revised manuscript. In order to keep the same nomenclature, "ROAC" and "smart polymerization" were replaced by "ROCOP" and "switchable polymerization", respectively. The revisions have been marked in red color in the revised manuscript and revised Supporting Information.

After revision

Page 2 in the revised manuscript.

Figure 1. One-step synthesis of multiblock polymers from a monomer mixture.

Figure 3. The synthesis of multiblock polymers with various sequences by the smart combination among different anhydride/epoxide ROCOP at 100 °C.

Page 5 in the revised manuscript.

Figure 4. Two different polymerization routes of the mixture of anhydride, epoxide, and L-LA: (a) the conventional route; (b) the new route.

Figure 8. The one-step synthesis of various core-shell type multiblock copolymers by the smart combination among different catalytic cycles at 80 °C.

Comment 2:

Polymerizations are not well controlled, $\bar{D} = 1.3 - 1.7$. This means there may be some transesterification side-reactions responsible for block scrambling? The authors should certainly prove this is not happening (^{13}C NMR can be useful here). perhaps some studies into ROP and ROCOP reactions can establish why the dispersity is high – e.g. initiation problems?

Response: Thank you very much for your careful checking and kind suggestions. I agree with your comments. The $\bar{D} = 1.3 - 1.7$ means that some transesterification side-reactions could occur during switchable polymerization. Therefore, we have tried to use ^{13}C NMR (including ^{13}C NMR spectra of the final product and crude aliquots withdrawn from the reaction system) to analyze whether transesterification side-reactions

occurs or not. Since final ^{13}C NMR spectra (Figure S48) includes many carbons, it was very difficult to identify the transesterification products by assigning the signals. Therefore, we used ^{13}C NMR spectra to monitor the polymerization processing, and we did not observe new carbonyl groups formed due to transesterification side-reactions during polymerization process (Figures S46 and S48). The result indicated that transesterification side-reactions should not occur. The SEC trace showed that dispersity remains narrow, and it became broad only when DPMA started to consume. This can be attributed to slow initiation caused by the very slow consumption of DPMA. Relevant discussion was added in the revised manuscript. The revisions were marked in red color.

Figure S46 was added on the page 43 in the revised Supporting Information.

Figure S46. The ^{13}C NMR (CDCl₃) spectra of crude aliquots withdrawn from the reaction system for monitoring the conversion of DGA, SA, NA, DPMA, L-LA, and the formation of resultant polymers (entry 8 in Table 1).

Figure S48. The ^{13}C NMR (CDCl_3) spectrum of the resultant polymer isolated from the mixture by precipitation (entry 8 in Table 1). Chemical shifts from the carbonyl groups of P(SA-*alt*-epoxide), P(NA-*alt*-epoxide), and P(DPMA-*alt*-epoxide) locate at 172.5 ppm ~ 171.5 ppm, and Chemical shifts from the carbonyl groups of P(DGA-*alt*-epoxide) and PLLA locate at 169.1 ppm ~ 170.0 ppm.⁴⁻⁶

Addition

Page 6 in the revised manuscript.

The formation process of the multiblock polymer was also confirmed by the evolution of the ^{13}C NMR spectrum (Figure S46). In particular, the ^{13}C NMR signals associated with carbonyl groups of DGA (carbon 1'), SA (carbon 7'), and NA (carbon 11') disappeared one by one, while the ^{13}C NMR signals associated with carbonyl groups of P(DGA-*b*-BO) (carbon 1), P(SA-*b*-BO) (carbon 7), and P(NA-*b*-BO) (carbon 11) segments appeared in succession. After 43.9% conversion of NA, the ^{13}C NMR signals associated with carbonyl groups of L-LA (carbon 18') also commenced decreasing, accompanied by the appearance of the ^{13}C NMR signals associated with carbonyl groups of the PLLA segment (carbon 18). During the process, the signals of carbonyl groups of DPMA (carbons 21' and 34') remained unchanged.

Page 6 in the revised manuscript.

The \mathcal{D} do not broaden until the complete consumption of L-LA (Figure 5d), and the increase of \mathcal{D} value was observed during the DPMA/epoxide ROCOP step. Due to the absence of new carbonyl groups caused by transesterification side-reactions during the polymerization process (Figures S46 and S48), we can reasonably deduce that very slow consumption of DPMA by ROCOP of DPMA/epoxides leads to slow initiation, which makes the \mathcal{D} become broad.

Comment 3:

Overall the polyester molar masses are low, <8 kDa, with Entry 4 at 2 kDa. What is the block DP for each segment and what is the confidence that all these blocks are present in the structure? Perhaps this could be addressed by stating the limits of catalyst loading and, therefore, molar mass?

Response: Thank you very much for your careful checking and kind suggestions. We used ^1H NMR to calculate the conversion of each monomer, and block DP of each segment can be calculated by conversion of these monomers. Evolution of SEC traces showed that molecular weight increased with increasing conversion of monomers, accompanied by a narrow, unimodal distribution. For example, for DGA/SA/NA/L-LA/DPMA polymerization system, after the conversion of one monomer (such as DGA) reached 100%, molecular weight still increased, which is because another monomer (such as SA) started to consume. Only one diffusion coefficient is observed in the DOSY of the final polymer, and this means that multiblock polymers were formed rather than blend. $M_{n, \text{SEC}}$ was obtained using polystyrene (PSt) as a standard, so the value seriously depends on the difference between the chemical structure of the synthesized polymer and PSt. Because the NMR based molecular weights for multi-block polymer from DGA/SA/NA/L-LA/DPMABO with PEG as initiator is not very precise, $M_{n, \text{MALS}}$ was obtained using size exclusion chromatography-multi-angle light scattering (SEC-MALS) and the value is 18.7 kDa, which is close to $M_{n, \text{th}}$ (16.5 kDa). Therefore, absolute molar mass is higher than 8 kDa. The polyester from Entry 4 molar mass is very low (2 kDa), which could be attributed to the transesterification reactions at 100 °C, but the side reaction can be suppressed by decreasing the temperature from 100 °C to 80 °C (entry 6 in Table 1, $\mathcal{D} = 1.35$, $M_{n, \text{SEC}} = 4.2$ kDa, Figure S37)

Figure S37. The SEC (THF) trace of the resultant triblock polymers from DPMA/L-LA/BO mixture (entry 6 in Table 1).

Addition:

Page 11 in the revised Supporting Information.

The polymerization of DGA/NA/EGE

Calculation method for degree of polymerization (DP) of each block:

$$DP_{P(DGA-alt-EGE)} = 12.5 \times \text{conv. of DGA} = 12.5 \text{ per block}$$

$$DP_{P(NA-alt-EGE)} = (12.5 \times \text{conv. of NA})/2 = 5.8 \text{ per block}$$

Page 22 in the revised Supporting Information.

The polymerization of DGA/SA/NA/DPMA/BO

Calculation method for degree of polymerization (DP) of each block:

$$DP_{P(DGA-alt-BO)} = 12.5 \times \text{conv. of DGA} = 12.5 \text{ per block}$$

$$DP_{P(SA-alt-BO)} = (12.5 \times \text{conv. of SA})/2 = 6.3 \text{ per block}$$

$$DP_{P(NA-alt-BO)} = (12.5 \times \text{conv. of NA})/2 = 6.3 \text{ per block}$$

$$DP_{P(DPMA-alt-BO)} = (6.25 \times \text{conv. of DPMA})/2 = 2.7 \text{ per block}$$

Page 26 in the revised Supporting Information.

The polymerization of DGA/SA/NA/EGE

Calculation method for degree of polymerization (DP) of each block:

$$DP_{P(\text{DGA-}alt\text{-EGE})} = 12.5 \times \text{conv. of DGA} = 12.5 \text{ per block}$$

$$DP_{P(\text{SA-}alt\text{-EGE})} = (12.5 \times \text{conv. of SA})/2 = 6.3 \text{ per block}$$

$$DP_{P(\text{NA-}alt\text{-EGE})} = (12.5 \times \text{conv. of NA})/2 = 4.5 \text{ per block}$$

$$P(\text{NA-}alt\text{-EGE})_{4.5}\text{-}b\text{-}P(\text{SA-}alt\text{-EGE})_{6.3}\text{-}b\text{-}P(\text{DGA-}alt\text{-EGE})_{12.5}\text{-}b\text{-}P(\text{SA-}alt\text{-EGE})_{6.3}\text{-}b\text{-}P(\text{NA-}alt\text{-EGE})_{4.5}$$

Page 30 in the revised Supporting Information.

The polymerization of L-LA/DPMA/EGE at 100 °C

Calculation method for degree of polymerization (DP) of each block:

$$DP_{\text{PLLA}} = 25 \times \text{conv. of L-LA} = 25 \text{ per block}$$

$$DP_{P(\text{DPMA-}alt\text{-EGE})} = (12.5 \times \text{conv. of DPMA})/2 = 5 \text{ per block}$$

$$P(\text{DPMA-}alt\text{-EGE})_5\text{-}b\text{-}PLLA_{25}\text{-}b\text{-}P(\text{DPMA-}alt\text{-EGE})_5$$

Page 36 in the revised Supporting Information.

The polymerization of L-LA/DPMA/BO at 80 °C

Calculation method for degree of polymerization (DP) of each block:

$$DP_{\text{PLLA}} = 25 \times \text{conv. of L-LA} = 25 \text{ per block}$$

$$DP_{P(\text{DPMA-}alt\text{-BO})} = (12.5 \times \text{conv. of DPMA})/2 = 2.8 \text{ per block}$$

$$P(\text{DPMA-}alt\text{-BO})_{2.8}\text{-}b\text{-}PLLA_{25}\text{-}b\text{-}P(\text{DPMA-}alt\text{-BO})_{2.8}$$

Page 42 and 43 in the revised Supporting Information.

The polymerization of DGA/SA/L-LA/DPMA/BO at 80 °C and 100 °C

Calculation method for degree of polymerization (DP) of each block (reaction at 100 °C)

$$DP_{P(\text{DGA-}alt\text{-BO})} = 12.5 \times \text{conv. of DGA} = 12.5 \text{ per block}$$

$$DP_{P(\text{SA-}alt\text{-BO})} = (12.5 \times \text{conv. of SA})/2 = 6.3 \text{ per block}$$

$$DP_{P(\text{NA-}alt\text{-BO})} = (12.5 \times \text{conv. of NA})/2 \times 0.287 = 1.8 \text{ per block}$$

$$DP_{\text{PLLA}} = (37.5 \times \text{conv. of L-LA})/2 \times 0.344 = 6.5 \text{ per block}$$

$$DP_{\text{P(NA-}i alt\text{-BO-grad-L-LA)}} = (12.5 \times \text{conv. of NA})/2 \times 0.713 + (37.5 \times \text{conv. of L-LA})/2 \times 0.656 = 16.8 \text{ per block}$$

$$DP_{\text{P(DPMA-}i alt\text{-BO)}} = (6.25 \times \text{conv. of DPMA})/2 = 2.7 \text{ per block}$$

$$\text{P(DPMA-}i alt\text{-BO)}_{2.7-}b\text{-PLLA}_{6.5-}b\text{-P(NA-}i alt\text{-BO-grad-L-LA)}_{16.8-}b\text{-P(NA-}i alt\text{-BO)}_{1.8-}b\text{-P(SA-}i alt\text{-BO)}_{6.3-}b\text{-P(DGA-}i alt\text{-BO)}_{12.5-}b\text{-P(SA-}i alt\text{-BO)}_{6.3-}b\text{-P(NA-}i alt\text{-BO)}_{1.8-}b\text{-P(NA-}i alt\text{-BO-grad-L-LA)}_{16.8-}b\text{-PLLA}_{6.5-}b\text{-P(DPMA-}i alt\text{-BO)}_{2.7}$$

Calculation method for degree of polymerization (DP) of each block (reaction at 80 °C)

$$DP_{\text{P(DGA-}i alt\text{-BO)}} = 12.5 \times \text{conv. of DGA} = 12.5 \text{ per block}$$

$$DP_{\text{P(SA-}i alt\text{-BO)}} = (12.5 \times \text{conv. of SA})/2 = 6.3 \text{ per block}$$

$$DP_{\text{P(NA-}i alt\text{-BO)}} = (12.5 \times \text{conv. of NA})/2 \times 0.439 = 2.7 \text{ per block}$$

$$DP_{\text{PLLA}} = (37.5 \times \text{conv. of L-LA})/2 \times 0.422 = 7.9 \text{ per block}$$

$$DP_{\text{P(NA-}i alt\text{-BO-grad-L-LA)}} = (12.5 \times \text{conv. of NA})/2 \times 0.561 + (37.5 \times \text{conv. of L-LA})/2 \times 0.578 = 14.3 \text{ per block}$$

$$DP_{\text{P(DPMA-}i alt\text{-BO)}} = (6.25 \times \text{conv. of DPMA})/2 = 2.4 \text{ per block}$$

$$\text{P(DPMA-}i alt\text{-BO)}_{2.4-}b\text{-PLLA}_{7.9-}b\text{-P(NA-}i alt\text{-BO-grad-L-LA)}_{14.3-}b\text{-P(NA-}i alt\text{-BO)}_{2.7-}b\text{-P(SA-}i alt\text{-BO)}_{6.3-}b\text{-P(DGA-}i alt\text{-BO)}_{12.5-}b\text{-P(SA-}i alt\text{-BO)}_{6.3-}b\text{-P(NA-}i alt\text{-BO)}_{2.7-}b\text{-P(NA-}i alt\text{-BO-grad-L-LA)}_{14.3-}b\text{-PLLA}_{7.9-}b\text{-P(DPMA-}i alt\text{-BO)}_{2.4}$$

Page 47 in the revised Supporting Information.

The polymerization of DGA/SA/L-LA/DPMA/BO

Calculation method for degree of polymerization (DP) of each block

$$DP_{\text{P(DGA-}i alt\text{-BO)}} = 12.5 \times \text{conv. of DGA} = 12.5 \text{ per block}$$

$$DP_{\text{P(SA-}i alt\text{-BO)}} = (12.5 \times \text{conv. of SA})/2 = 6.3 \text{ per block}$$

$$DP_{\text{PLLA}} = (37.5 \times \text{conv. of L-LA})/2 = 18.8 \text{ per block}$$

$$DP_{\text{P(DPMA-}i alt\text{-BO)}} = (6.25 \times \text{conv. of DPMA})/2 = 1.3 \text{ per block}$$

$$\text{P(DPMA-}i alt\text{-BO)}_{1.3-}b\text{-PLLA}_{18.8-}b\text{-P(SA-}i alt\text{-BO)}_{6.3-}b\text{-P(DGA-}i alt\text{-BO)}_{12.5-}b\text{-P(SA-}i alt\text{-BO)}_{6.3-}b\text{-PLLA}_{18.8-}b\text{-P(DPMA-}i alt\text{-BO)}_{1.3}$$

Page 51 in the revised Supporting Information.

The polymerization of DGA/SA/NA/L-LA/DPMA/BO with PEG as an initiator

Calculation method for degree of polymerization (DP) of each block

$$DP_{P(DGA-alt-BO)} = 12.5 \times \text{conv. of DGA} = 6.3 \text{ per block}$$

$$DP_{P(SA-alt-BO)} = (12.5 \times \text{conv. of SA})/2 = 6.3 \text{ per block}$$

$$DP_{P(NA-alt-BO)} = (12.5 \times \text{conv. of NA})/2 \times 0.485 = 3.0 \text{ per block}$$

$$DP_{PLLA} = (37.5 \times \text{conv. of L-LA})/2 \times 0.427 = 8.0 \text{ per block}$$

$$DP_{P(NA-alt-BO-grad-L-LA)} = (12.5 \times \text{conv. of NA})/2 \times 0.515 + (37.5 \times \text{conv. of L-LA})/2 \times 0.573 = 14.0 \text{ per block}$$

$$DP_{P(DPMA-alt-BO)} = (6.25 \times \text{conv. of DPMA})/2 = 2.8 \text{ per block}$$

$$P(DPMA-alt-BO)_{2.8-b}-PLL A_{8.0-b}-P(NA-alt-BO-grad-L-LA)_{14.0-b}-P(NA-alt-BO)_{3.0-b}-P(SA-alt-BO)_{6.3-b}-P(DGA-alt-BO)_{6.3-b}-PEG-b-P(DGA-alt-BO)_{6.3-b}-P(SA-alt-BO)_{6.3-b}-P(NA-alt-BO)_{3.0-b}-P(NA-alt-BO-grad-L-LA)_{14.0-b}-PLL A_{8.0-b}-P(DPMA-alt-BO)_{2.8}$$

Page 59 in the revised Supporting Information.

The polymerization of TA/NA/L-LA/DPMA/BO

Calculation method for degree of polymerization (DP) of each monomer

$$DP_{TA} = 12.5 \times \text{conv. of TA} = 12.5$$

$$DP_{NA} = 12.5 \times \text{conv. of NA} = 12.5$$

$$DP_{L-LA} = 25 \times \text{conv. of L-LA} = 25$$

$$DP_{DPMA} = 6.25 \times \text{conv. of DPMA} = 3.4$$

Page 64 in the revised Supporting Information.

The polymerization of TA/NA/BO

Calculation method for degree of polymerization (DP) of each monomer

$$DP_{TA} = 12.5 \times \text{conv. of TA} = 12.5$$

$$DP_{NA} = 12.5 \times \text{conv. of NA} = 7.6$$

Page 67 in the revised Supporting Information.

The polymerization of TA/L-LA/DPMA/BO

Calculation method for degree of polymerization (DP) of each monomer

$$DP_{TA} = 12.5 \times \text{conv. of TA} = 12.5$$

$$DP_{L-LA} = 25 \times \text{conv. of L-LA} = 25$$

$$DP_{DPMA} = 6.25 \times \text{conv. of DPMA} = 3$$

Comment 4:

It's a bit difficult to see the motivation for making a star/dendritic copolymer and some proof of concept rheology or application testing would be really beneficial to illustrate why the community would adopt this approach.

Response: Thank you very much for your careful checking and kind suggestions. According to your comments, we have added comments regarding why the community would adopt this approach and the revision has been marked in red color in the revised manuscript.

Addition

Page 8. Core-shell-type multi-block copolymer has demonstrated several characteristics when compared with linear polymer, including a large population of terminal functional groups, lower melt viscosity, and better solubility.⁵⁰ Thus, it has attracted more and more attention from the scientific and engineering points of view. However, the synthesis of a core-shell-type multi-block copolymer is still limited to multi-step procedure.

50. Zhou, Y., Huang, W., Liu, J., Zhu, X., Yan, D. Self - assembly of hyperbranched polymers and its biomedical applications. *Advanced materials*, **22**, 4567-4590 (2010).

Reviewer: 3

General comments:

When multiple number of monomers differing in reactivity co-exist, the more reactive one gets consumed over the other(s). This is rather general especially in ionic polymerizations.

Here in this manuscript, the authors selected many monomers to accomplish a world record of the number of blocks. Although the concept itself is not new, I think it is still remarkable that they could come to this fine selection of monomers.

While I found the well-defined polymer structure is fascinating, it is unfortunate that the expressions used in this manuscript is somewhat misleading in many cases.

Thank you very much for your helpful comments! It is really helpful for us to improve our work. We have made revisions according to your comments.

Comment 1:

"The catalyst smartly switches" in Abstract, line 13. I don't think any catalyst is "smart" or "dull". It simply selects the monomers in the order of activation energy.

Response: Thank you very much for your careful checking and kind suggestions. We agree with your comment, and the sentence has been rephrased and marked in red color on page 1 in the revised manuscript.

Before revision

Switchable polymerization holds considerable potential for simulating the molecular precision of natural biopolymers, such as nucleic acids or proteins, to synthesize highly sequence-controlled macromolecules. To date, this method has been limited to three-component systems, which enables the straightforward synthesis of multiblock polymers with less than five blocks. Herein, we report a self-switchable polymerization enabled by simple alkali metal carboxylate catalysts that directly polymerize six-component mixtures into multiblock polymers consisting of up to 11 blocks. The catalyst spontaneously and smartly switches among five catalytic cycles, involving four anhydride/epoxide ring-opening copolymerizations and one L-lactide ring-opening

polymerization, creating a one-step synthetic pathway. Following this smart catalysis, reasonable combinations of different catalytic cycles allow the direct preparation of diverse, sequence-controlled, multiblock copolymers even containing various hyperbranched architectures. This method shows considerable promise in the synthesis of sequentially and architecturally complex polymers, with high monomer sequence control that provides the potential for designing materials.

After revision

Switchable polymerization holds considerable potential for the synthesis of highly sequence-controlled multiblock. To date, this method has been limited to three-component systems, which enables the straightforward synthesis of multiblock polymers with less than five blocks. Herein, we report a self-switchable polymerization enabled by simple alkali metal carboxylate catalysts that directly polymerize six-component mixtures into multiblock polymers consisting of up to 11 blocks. Without external trigger, the catalyst polymerization spontaneously connects five catalytic cycles in an orderly manner, involving four anhydride/epoxide ring-opening copolymerizations and one L-lactide ring-opening polymerization, creating a one-step synthetic pathway. Following this autotandem catalysis, reasonable combinations of different catalytic cycles allow the direct preparation of diverse, sequence-controlled, multiblock copolymers even containing various hyperbranched architectures. This method shows considerable promise in the synthesis of sequentially and architecturally complex polymers, with high monomer sequence control that provides the potential for designing materials.

Comment 2:

Also, the word "switch" is not suitable. Simply, once one monomer is consumed, the next monomer starts to react.

Response: Thank you very much for your careful checking and kind suggestions. According to your comments, appropriate expression has been used to describe the polymerization process, and the revision has been marked in red color in the revised manuscript and revised Supporting Information.

Comment 3:

The introduction is quite misleading. The polymer obtained in this work still varies in the unit numbers of each

block, which is contrastive to the perfectly controlled sequence of biopolymers such as nucleic acids or proteins. Given that the method presented here inherently never reach the level of control of biopolymers, the introduction should be completely rewritten without the unnecessary comparisons.

Response: Thank you very much for your careful checking and kind suggestions. According to your comments, we have reedited the contents in the introduction, and the revisions were marked in red color on page 1 in the revised manuscript.

Before revision

Page 1. Advanced biological systems have naturally evolved, thereby providing a green, smart pathway for preparing diverse, sequence-defined biopolymers, such as DNA, RNA, and proteins. Precise monomer sequence regulation plays a vital role in biology and is a prerequisite for crucial features, such as heredity, self-replication, complex self-assembly, and molecular recognition.¹⁻³ Inspired by biology, chemists have developed various synthetic methods, including “click” reactions,^{4,5} sequential monomer addition,⁶⁻⁸ and solid-phase synthesis,⁹⁻¹¹ to control monomer sequences in synthetic macromolecules. However, these strategies are hampered by disadvantages, such as being extremely complex and time-consuming, as well as iterative monomer attachment/deprotection. This increases costs and often leads to poor yields, thereby making it challenging to expand their application.¹²⁻¹⁴

“Switchable polymerization” has been exploited for the spontaneous, selective transformation of a monomer mixture into a sequence-controlled block copolymer in one synthetic step, showing similar features to those in nature and thereby overcoming the disadvantages of conventional procedures.

After revision

Page 1. Copolymers are long macromolecular chains composed of at least two monomers of different chemical natures. High monomer sequence regulation enables effective control of structure-property relations of copolymers so that precise sequence-controlled polymers may be endowed with novel properties or functions.¹⁻³ In this context, considerable efforts have been made on the development of various synthetic methods, including “click” reactions,^{4,5} sequential monomer addition,⁶⁻⁸ and solid-phase synthesis.⁹⁻¹¹ Although these strategies have made significant progress for the synthesis of sequence-controlled block

polymers, they are hampered by disadvantages, such as being extremely complex and time-consuming, as well as iterative monomer attachment/deprotection. This increases costs and often leads to poor yields, thereby making it challenging to expand their application.¹²⁻¹⁴

“Switchable polymerization” has been exploited for the spontaneous, selective transformation of a monomer mixture into a sequence-controlled block copolymer in one synthetic step, thereby overcoming the disadvantages of conventional procedures.

Comment 4:

Time course profile of SEC peak should be added in the Supporting Information (Not only the final product).

Response: Thank you very much for your careful checking and kind suggestions. Evolution of SEC traces have been added in the revised Supporting Information.

After revision

Page 14 in the Supported Information.

The polymerization of DGA/NA/EGE

Figure S7. The SEC (THF) trace of the resultant triblock polymer (entry 1 in Table 1)

Page 23 in the Supported Information.

The polymerization of DGA/SA/NA/DPMA/BO

Figure S20. The SEC (THF) trace of the resultant heptablock polymers (entry 2 in Table 1).

Page 27 in the Supported Information.

The polymerization of DGA/SA/NA/EGE

Figure S25. The SEC (THF) trace of the resultant pentablock polymers (entry 3 in Table 1).

Page 48 in the Supported Information.

The polymerization of DGA/SA/L-LA/DPMA/BO

Figure S53. The SEC (THF) trace of the resultant heptablock polymers (entry 9 in Table 1).

Page 61 in the Supported Information.

The polymerization of TA/NA/L-LA/DPMA/BO

Figure S70. The SEC (THF) trace of the resultant polymers (entry 11 in Table 1).

Page 65 in the Supported Information.

The polymerization of TA/NA

Figure S75. The SEC (THF) trace of the resultant polymers (entry 12 in Table 1).

Page 68 in the Supported Information.

The polymerization of TA/L-LA/DPMA/BO

Figure S79. The SEC (THF) trace of the resultant polymers (entry 13 in Table 1).

Comment 5:

In the initiating group, RO- oxygen is missing in Figure 1.

Response: Thank you very much for your careful checking and kind suggestions. We have modified Figure 1, and the correct format have shown on page 2 in the revised manuscript.

After revision

Page 2 in the revised manuscript.

Figure 1. One-step synthesis of multiblock polymers from a monomer mixture. (a) Previous strategies are only applicable to three-component monomer mixture. (b) The strategy presented in this paper provides multiblock polymers from six-component monomer mixture. L-LA = L-lactide, DGA = diglycolic anhydride, NA = 5-norbornene-endo-2,3-dicarboxylic anhydride, EGE = ethyl glycidyl ether, SA = succinic anhydride, DPMA = rac-cis-endo-1-isopropyl-4-methyl-bicyclo[2.2.2]oct-5-ene-2,3-dicarboxylic anhydride, BO = 1,2-butylene oxide, ROCOP = ring-opening copolymerization, ROP = ring-opening polymerization, PEG2000 = polyethylene glycol (molecular weight = 2 kDa).

REVIEWER COMMENTS

Reviewer #1 (Remarks to the Author):

The answers/corrections made by the authors, according to my comments, are satisfactory. However, I will agree with the publication of this MS in Nature Communications on the following condition:

The authors should clearly state in the Abstract and Conclusions that the synthesized multiblock polymers have relatively high PDI and tapered sequences between the blocks (not perfect block copolymers).

Reviewer #1 also looked over the authors' comments to reviewer #3's suggestion:

After reading the authors' responses to the comments, I am skeptical whether this MS should be published in a high-level scientific Journal such as Nature Communications.

The authors respond to the critical comments on "well-defined polymer structure" or "smart catalyst" (general comment and comments 1, 2, and 3) by accepting them without stating the reasons why they used this terminology. In addition, the polydispersity indexes \bar{M}_w/\bar{M}_n calculated from the corresponding SEC traces (comment 4) are not consistent. For example, in Figure S20, the black SEC trace corresponds to $\bar{M}_w/\bar{M}_n=1.28$ and the pink to $\bar{M}_w/\bar{M}_n=1.30$! Impossible.

Reviewer #2 (Remarks to the Author):

the revisions are very thorough - thank you. There's just one area not addressed - the Title - probably it would be wise to remove the term 'smart'

We appreciate the comments from the editor and the reviewers. The following is our response to the comments.

Reviewer: 1

General comments: The answers/corrections made by the authors, according to my comments, are satisfactory. However, I will agree with the publication of this MS in Nature Communications on the following condition.

Response: Thank you very much for your helpful comments! It is really helpful for us to improve our work. We have made revisions according to your comments.

Comment 1:

The authors should clearly state in the Abstract and Conclusions that the synthesized multiblock polymers have relatively high PDI and tapered sequences between the blocks (not perfect block copolymers).

Response: Thank you very much for your careful checking and kind suggestions. As you said, for the DGA/SA/NA/L-LA/DPMA and TA/NA/L-LA/DPMA mixture system, the obvious tapered region would form in these systems because the reactivity difference between L-LA and NA is very small. However, the nearly perfect multiblock copolymers can be also prepared by rational combination of monomer mixture system. Therefore, the formation of tapered region should be described in the specific polymerization system, and according to your comments, we have commented regarding this issue clearly in the main text. Due to the 150-words limit for the abstract, it is better to focus on stating the ability of the present polymerization system to selectively enchain mixtures of 6 different monomers using different mechanisms, presenting its advance in the field and potential for designing materials. According to your comment, we clearly stated in the conclusions that some of the synthesized multiblock polymers show relatively high PDI and the tapered region between the PLLA and P(NA-*alt*-BO) blocks. The revisions were marked in red color on page 10 in the revised manuscript.

Before revision

In conclusion, a versatile, direct, one-step synthesis of a well-defined multiblock polyester of up to 11 blocks from a six-component mixture was demonstrated. The alkali metal carboxylate catalyst spontaneously connected five catalytic cycles, involving four cyclic anhydride/epoxide ROCOPs and an L-LA ROP. Control over the monomer incorporation sequence based on reactivity ratio of these monomers (DGA \gg SA \gg NA $>$ L-LA \gg DPMA and TA \gg NA $>$ L-LA \gg DPMA) rendered the switchable polymerization similar to ideal examples in nature, allowing the synthesis of different sequence-controlled multiblock copolymers even containing various hyperbranched architectures. A notable advantage of this method was the ability to freely manipulate the polymerization order between anhydride/epoxide ROCOP and LA ROP, creating a more flexible polymerization pathway. Thus, the simple, smart, sequence-controlled polymerization yielded tailored functional materials for high-value emerging applications, such as data storage, anti-counterfeiting technologies, microelectronics, and nanomedicine. However, the essential factor that determined the reactivity differences between these monomers is yet to be determined. Ongoing studies are focusing on revealing this factor and extending the applicability to a large library of structurally and functionally diverse cyclic anhydrides, epoxides, and cyclic esters.

After revision

In conclusion, a versatile, direct, one-step synthesis of a sequence-controlled multiblock polyester of up to 11 blocks from a six-component mixture was demonstrated. The alkali metal carboxylate catalyst spontaneously connected five catalytic cycles, involving four cyclic anhydride/epoxide ROCOPs and an L-LA ROP. Control over the monomer incorporation sequence based on reactivity ratio of these monomers (DGA \gg SA \gg NA $>$ L-LA \gg DPMA and TA \gg NA $>$ L-LA \gg DPMA) rendered the switchable polymerization similar to ideal examples in nature, allowing the synthesis of different sequence-controlled multiblock copolymers even containing various hyperbranched architectures with the relative broad \bar{D} (1.23–1.71). Although the obvious tapered region was formed when combining L-LA and NA because of their similar reactivity, nearly perfect multiblock polymers can be obtained by rational combination of different polymerization cycles. A notable advantage of this method was the ability to freely manipulate the polymerization order between anhydride/epoxide ROCOP and L-LA ROP, creating a more flexible polymerization pathway. Thus, the simple, smart, sequence-controlled polymerization yielded tailored functional materials for high-value emerging

applications, such as data storage, anti-counterfeiting technologies, microelectronics, and nanomedicine. However, the essential factor that determined the reactivity differences between these monomers is yet to be determined. Ongoing studies are focusing on revealing this factor and extending the applicability to a large library of structurally and functionally diverse cyclic anhydrides, epoxides, and cyclic esters.

Comment 2:

Reviewer #1 also looked over the authors' comments to reviewer #3's suggestion:

After reading the authors' responses to the comments, I am skeptical whether this MS should be published in a high-level scientific Journal such as Nature Communications. The authors respond to the critical comments on "well-defined polymer structure" or "smart catalyst" (general comment and comments 1, 2, and 3) by accepting them without stating the reasons why they used this terminology.

Response: Thank you very much for your careful checking and kind suggestions. Although we accepted the reviewer's comments without any reason, we revised the manuscript after careful consideration. Initially, we wanted to use the term "smart polymerization" to define the present polymerization system, because sequence-controlled multiblock polymer can be directly synthesized from the 6 monomers mixture without any redundant process. Based on this, we thought that precision synthesis of biomacromolecules (such as DNA) is suited to illustrate our polymerization system that can provide the smart way for simulating the molecular precision of natural biopolymers synthesis. However, based on the reviewer's comments, we gave up some terminologies after careful consideration, including the "smart", "smart catalyst or polymerization", and "smartly switch" for the following reasons: 1) Our one-step synthesis of multiblock polymer is realized by a single catalyst connecting different catalytic cycles, but also highly depends on reactivity difference among the monomers. Therefore, it is unable to indicate that catalyst has the feature of "smart". 2) Although "smart polymerization" is a new and exciting concept, it is somewhat confusing so that it is unable to describe clearly the polymerization process, compared with the "self-switchable polymerization". 3) The "switchable catalyst" concept was first proposed by Williams and co-workers. The switch catalysis concept is now proven to involve a single catalyst that switches at least two different mechanisms such as epoxide/anhydride ROCOP and lactone ROP. Therefore, the term "switch" can be used for linking two polymerization mechanisms, while using the term "switch" to describe the orderly monomers consumption could lead to misunderstanding. Based on this, we give up to use the term "switch" in some places of this manuscript.

On the other hand, we did not completely give up the terminology of “well-defined multiblock polymer”. Although the polymer obtained in this work is far from the level of control in biopolymers, we can synthesize well-defined multiblock polymer by the rational design of polymerization system. The sequence of the obtained polymer can be controlled and its structure can be quantified.

Comment 3:

In addition, the polydispersity indexes \bar{D} s calculated from the corresponding SEC traces (comment 4) are not consistent. For example, in Figure S20, the black SEC trace corresponds to \bar{D} =1.28 and the pink to \bar{D} =1.30! Impossible.

Response: Thank you very much for your careful checking and kind suggestions. As you said, the pink SEC trace in Figure S20 looks broader than the black one so that we also worried that these data are incorrect. Therefore, we have rechecked SEC results and calculated the polydispersity indexes \bar{D} s again for all cases. For Figure S20, the black and pink SEC traces correspond to \bar{D} =1.277 and \bar{D} =1.309, respectively, as shown in the calculation results from the SEC data processing software (Figure R1), which are almost the same as the previous values. For the molecular weight of the obtained polymer, there is a small difference between this value (2.7 kDa at 5.0 h and 7.5 kDa at 73.0 h) and previous one (2.8 kDa at 5.0 h (black) and 7.4 kDa at 73.0 h (pink)), and the small difference is caused by difference in the calculation areas. This small difference was occurred in the other polymerization systems, as shown in Figures R2 and R3, respectively. Therefore, we adopted the results calculated at this time in the revised manuscript and the revisions were marked in red color on page 3 in the revised manuscript and the revised Supporting Information.

Figure R1. Original calculation results (screen shot of the data processing software) for the SEC data in Figure S20. (a) SEC trace of obtained polymer at 5.0 h (black line); (b) SEC trace of obtained polymer at 73.0 h (pink line).

After revision

Figure S20. The SEC (THF) trace of the resultant heptablock polymers (entry 2 in Table 1).

Page 27 in the Supporting Information.

The polymerization of DGA/SA/NA/EGE

Figure R2. Original calculation results (screen shot of the data processing software) for the SEC data in Figure S25. (a) SEC trace of obtained polymer at 3.0 h (black line); (b) SEC trace of obtained polymer at 5.0 h (red line); (c) SEC trace of obtained polymer at 7.0 h (blue line).

After revision

Figure S25. The SEC (THF) trace of the resultant pentablock polymers (entry 3 in Table 1).

Figure R3. Original calculation results (screen shot of the data processing software) for the SEC data in Figure S53. (a) SEC trace of obtained polymer at 3.0 h (black line); (b) SEC trace of obtained polymer at 7.0 h (red line); (c) SEC trace of obtained polymer at 22.7 h (blue line); (d) SEC trace of obtained polymer at 97.0 h (pink line).

After revision

Figure S53. The SEC (THF) trace of the resultant heptablock polymers (entry 9 in Table 1).

Reviewer #2 (Remarks to the Author):

the revisions are very thorough - thank you. There's just one area not addressed - the Title - probably it would be wise to remove the term 'smart'

Response: Thank you very much for your helpful comments! It is really helpful for us to improve our work. Based on your and other reviewers' comments, we have reconsidered whether "smart" is suitable or not. Initially, we wanted to use the "smart polymerization" to define the present polymerization system, because sequence-controlled multiblock polymer can be directly synthesized from the six monomers mixture without any redundant process. However, the polymer obtained in this work is far from the level of control in biopolymers. In addition, "smart polymerization" may be somewhat confusing. In order to objectively and scientifically present the research, we have decided to remove the term "smart". The revisions were marked in red color on page 1 in the revised manuscript.

REVIEWERS' COMMENTS

Reviewer #1 (Remarks to the Author):

The authors took into account all my concerns and addressed or corrected their MS. In my opinion, the MS is now ready for publication.

REVIEWERS' COMMENTS

Reviewer #1 (Remarks to the Author):

The authors took into account all my concerns and addressed or corrected their MS. In my opinion, the MS is now ready for publication.

Response: Thank you very much for your helpful comments! It is really helpful for us to improve our work.